# The Application of Prodrugs as a Tool to Enhance the Properties of Nucleoside Reverse Transcriptase Inhibitors

**DOI:** 10.3390/v15112234

**Published:** 2023-11-09

**Authors:** Lívia da Rocha Fernandes, Juliana Romano Lopes, Andressa Francielli Bonjorno, João Lucas Bruno Prates, Cauê Benito Scarim, Jean Leandro Dos Santos

**Affiliations:** School of Pharmaceutical Sciences, São Paulo State University (UNESP), Araraquara 14800-903, Brazil; livia.fernandes@unesp.br (L.d.R.F.); jromanolopes@gmail.com (J.R.L.); andressa.bonjorno@unesp.br (A.F.B.); joao.prates@unesp.br (J.L.B.P.); caue.scarim@unesp.br (C.B.S.)

**Keywords:** prodrugs, HIV, AIDS, nucleoside reverse transcriptase inhibitors (NRTIs), codrugs, new drugs, medicinal chemistry

## Abstract

Antiretroviral Therapy (ART) is an effective treatment for human immunodeficiency virus (HIV) which has transformed the highly lethal disease, acquired immunodeficiency syndrome (AIDS), into a chronic and manageable condition. However, better methods need to be developed for enhancing patient access and adherence to therapy and for improving treatment in the long term to reduce adverse effects. From the perspective of drug discovery, one promising strategy is the development of anti-HIV prodrugs. This approach aims to enhance the efficacy and safety of treatment, promoting the development of more appropriate and convenient systems for patients. In this review, we discussed the use of the prodrug approach for HIV antiviral agents and emphasized nucleoside reverse transcriptase inhibitors. We comprehensively described various strategies that are used to enhance factors such as water solubility, bioavailability, pharmacokinetic parameters, permeability across biological membranes, chemical stability, drug delivery to specific sites/organs, and tolerability. These strategies might help researchers conduct better studies in this field. We also reported successful examples from the primary therapeutic classes while discussing the advantages and limitations. In this review, we highlighted the key trends in the application of the prodrug approach for treating HIV/AIDS.

## 1. Introduction

Considerable advancements in the reduction of mortality and morbidity related to HIV infection were only possible after the development of ART [1]. However, social stigma, poor access to treatment, the persistence of a chronic inflammatory state and HIV latent reservoirs, the emergence of resistant strains, the lack of patient immune response, and drug-related adverse effects are major problems that need to be addressed [1]. In 2021, the WHO estimated that 38.4 million people are living with HIV; only 75% of them have access to ART [2].

More than 30 antiretroviral drugs belonging to different therapeutic classes have been approved by the Food and Drug Administration (FDA) [3]. These drugs can inhibit the main steps of the HIV viral cycle [3]. The capsid inhibitor (i.e., lenacapavir) represents the most recent drug category for these treatment options [3]. New therapeutic strategies for treating HIV infections that can be considered “emerging perspectives”, such as the allogenic transplant of stem cells from CCR5Δ32/Δ32 donors, ‘kick and kill’, ‘block and lock’, therapeutic vaccination, and gene editing (i.e., zinc finger nuclease and CRISPR-Cas-9), have emerged [1,2,3]. 

ART, in which at least two to three different antiretroviral drugs are administered, can be used to reduce the viral load into “non-detectable” levels in the plasma [3]. This treatment strategy has greatly increased the life expectancy and quality of life of patients [3,4]. However, challenges, including the emergence of HIV-resistant strains, coinfections, and long-term adverse effects (e.g., poor tolerability, drug interactions, and toxicity), are factors that increase the chances of poor patient compliance. Safe and convenient regimens for nonoccupational postexposure prophylaxis (nPEP) and mainly for preexposure prophylaxis (PrEP) are key factors that influence patient adherence to treatment [3,4].

The lack of treatment adherence is a complex phenomenon influenced by several factors, including intrinsic factors related to patients (e.g., psychosocial, socioeconomic), medical conditions, and pharmacological therapy [5]. A greater pill burden, complex posology schemes involving high intake frequency, the use of some class of therapeutic agents (e.g., protease inhibitors), adverse effects, and toxicity decrease adherence to ART [5]. The lack of adherence to treatment and the interruption of therapy can lead to disease progression and viral rebound due to the transcription of viral genomes silenced in CD4+ T cells and macrophage reservoirs found in various anatomical sites, such as the central nervous system and lymphatic system [1,6].

Among the strategies to overcome the limitations of antiviral drugs, prodrugs might be administered to modulate pharmacokinetic properties. This method can help maintain effective drug concentrations through a single dose/day regimen, reduce toxicity and adverse effects, and enhance the efficacy of drug delivery to target cells/organs [6,7,8]. Prodrugs are compounds with little or no pharmacological activity; however, chemical or enzymatic activation releases the active metabolite from prodrugs. Prodrugs comprise approximately 10% of all marketed drugs [8]. They are commonly used in drug design to improve water-solubility, increase lipophilicity, permeability, and oral absorption, promote parental and topical use, reduce toxicity and undesirable organoleptic properties, and decrease metabolic instability to accurately deliver the drug to the target site and prolong the duration of action [8,9,10]. 

This review covers recent advancements in the prodrug strategy linked to the creation of new HIV antiviral agents reported in the literature from 2010 to 2022. We emphasize key studies that have contributed to enhancing drug properties and delve into the most successful examples within various therapeutic classes.

## 2. Nucleoside Reverse Transcriptase Inhibitors—NRTIs

The synthesis of viral double-stranded DNA from viral RNA is performed by HIV reverse transcriptase (RT) [11]. This asymmetric heterodimer enzyme has two subunits (p51 and p66) derived from a Gag-Pol polyprotein [11]. It exhibits two enzymatic activities, including DNA polymerase and RNase H, which act synergistically to convert RNA into DNA. The DNA polymerase domain present in p66 is made of four distinct subdomains: fingers (residues 1–85; 118–155), palm (86–117; 156–236), thumb (237–318), and the connection (319–426). The nucleic acid is accommodated in the p66 thumb due to the contribution of αH and αI helices. The polymerase active site in the palm subunit of p66 exhibits three carboxylates (D110, D185, and D186) responsible for binding to the Mg^2+^ ion [11]. These carboxylates (D185 and D186) are described as a highly conserved YXDD motif that is found in all retroviruses; in HIV RT, X represents methionine. Additional conserved sites include K65 and R72, which are involved in the interactions with the phosphates of deoxynucleoside triphosphate (dNTP), Y115, which interacts with the deoxyribose ring, and Q151, which interacts with the 3′-OH of dNTPs [11].

Many FDA-approved drugs are inhibitors of HIV-1 reverse transcriptase, which belong to two different classes, including nucleoside RT inhibitors (NRTIs) and nonnucleoside RT inhibitors (NNRTIs) [12,13]. Examples of FDA-approved drugs include abacavir (ABC), didanosine (ddI), emtricitabine, lamivudine (3TC), stavudine (D4T), and tenofovir. All these drugs were developed through molecular modifications of natural nucleosides, and all of them involve the removal of the ribosyl 3′-hydroxyl group to induce chain termination blockade. Although this class is extremely important for controlling the replication of viruses, some issues in the common pharmacokinetic properties were found in NRTIs, which encouraged the use of prodrugs to optimize these effects [14,15]. Here, we discuss how the prodrug approach can improve the properties of NRTI-based compounds. We also present examples of applications and perspectives. 

### 2.1. Zidovudine Prodrugs

Zidovudine (3′-azido-3′-deoxythymidine; AZT) is a nucleoside RT inhibitor (NRTI) that contains an azido group (N_3_-R) at the third carbon of the dideoxyribose subunit [16]. Although AZT was synthesized in 1964 by Horwitz et al., its effect as an anti-HIV drug was shown in 1986 by Yarchoan et al. AZT was the first approved HIV-1 drug; however, it exhibits serious adverse effects, such as bone marrow toxicity, neutropenia, macrocytic anemia, and granulocytopenia. Subsequently, safer and more effective therapeutic alternatives were introduced [17]. Despite these limitations, this drug can be used as a prototype to investigate the use of the prodrug strategy [17]. In this section, we present some examples of zidovudine prodrugs.

The synthesis of AZT-glycerolipid prodrugs showed that this method can be used to develop liponucleosides. It can mimic the metabolism of natural triglycerides and can be directly transported to the lymphatic system without undergoing primary metabolism; thus, its bioavailability is high. Shastina et al. (2013) synthesized molecules that act as AZT prodrugs, with the drug conjugated to functional phosphorus groups. In studies on chemical stability, the phosphodiester derivative (compound (**1**), Figure 1) exhibited a half-life (t_1/2_) of 20 h at pH 7.3, a t_1/2_ of > 20 h at pH 8.5, and a t_1/2_ of 8 h at pH 9.5 [18]. The maximum antiretroviral activity of compound (**1**) was 94.1% at 10 µg/mL, whereas the maximum antiretroviral activity of AZT was 96.9% at 1 µg/mL [18]. Compound (**1**) also induced a decrease in the level of p24 antigen by 88.1% at 10 µg/mL. In toxicity studies, the compound showed a cellular viability of 77.67% at 10 µg/mL [18].

A new dinucleoside phosphonate of AZT, β-L-2′,3′-didesoxy-3′-tiacitidine (3TC), was synthesized by Solyev et al. (2014) with the aim of investigating the prodrug approach. The homodimer 3-azido-3-deoxythymidin-5-*O*-yl fluoromethylphosphonate (**2**) (Figure 1) was found to be the most effective compound against HIV, with a half maximal effective concentration (EC_50_) value of 0.015 µM in CEM-SS cells [19]. However, it was five times more toxic than the parental drug (AZT), with a 50% cytotoxic concentration (CC_50_) value of 29 µM, which indicated that it had a low selectivity index [SI (CC_50_/EC_50_)] of 1933 [19]. In this assay, AZT exhibited an EC_50_ value of 0.037 µM, a CC_50_ value of 142 µM, and an SI of 3837. Studies on the stability and hydrolysis of the (**2**) prodrug in human blood serum showed that it underwent fast hydrolysis and had a t_1/2_ of 0.78 h, i.e., it was slightly less stable than the other compounds evaluated in this work [19]. A mixture of AZT and AZT monophosphonate products was found in the hydrolysis assay [19].

McGuigan et al. (2013) prepared and evaluated AZT phosphorodiamidate prodrugs, prodrug (**3**) being a phosphodiamidate containing esterified L-alanine and a CH_2_tBu radical (Figure 1). It exhibited the best antiviral activity, as it could efficiently deliver the monophosphate form of the parental AZT nucleoside into cells. This prodrug (**3**) also had a high cellular uptake due to its high lipophilicity. The results of the cytotoxicity evaluation showed that prodrug (**3**) had A CC_50_ value of 75 µM, which was higher than that of AZT [20]. Its chemical and metabolic stability was evaluated using Trizma buffer (pH = 7.6) in the presence of carboxypeptidase Y, an enzyme that metabolizes the phosphorodiamidate subunit [20]. The study was conducted using the NMR technique. The results showed that the phosphorodiamidate subunit was susceptible to hydrolysis by carboxypeptidase Y. Prodrug (**3**) exhibited anti-HIV activity with an EC_50_ value of 0.0083 µM (HIV-1) and 0.013 µM (HIV-2) [20].

Kock et al. (2014) developed macromolecular azidothymidine polymeric prodrugs containing monomers with a self-immolative linker, which were evaluated against HIV-1. The presence of disulfide contributed to the intracellular release of AZT. The self-immolative linker spontaneously decomposed after the cleavage of the trigger, releasing the parental drug (Figure 1, (**4**)). After synthesizing the monomer, this polymeric prodrug was prepared through the controlled radical polymerization technique (RAFT). The macromolecular prodrugs were obtained through the treatment of monomers with *N*-(2-hydroxypropyl)methacrylamide (HPMA). The synthesized prodrug (**4**) exhibited an average molar mass ranging from 10 to 20 kDa. For them, it was observed constituting up to 20 mol% of the parental drug (AZT). The prodrug (**4**) also released AZT when it was exposed to 5 mM glutathione (GSH); the rate of release of AZT was high (t_1/2_ < 30 min). HIV infectivity was determined after mammalian cells were incubated with the polymers for 24 h, and the level of the virus was quantified after 48 h. The macromolecular AZT polymeric prodrug decreased HIV-1 infectivity by up to 80%. To evaluate the effect of the self-immolative linker, the researchers also prepared and evaluated ester analogs. The thiol-responsive prodrugs (**4**) were >10-fold more efficacious than their ester-based counterparts [21].

Bolaamphiphiles are amphiphilic compounds that contain a hydrophobic spacer linked to two hydrophilic compounds. Such structures tend to form monolayers and are commonly used as carriers of drugs and genes. An asymmetric-bolaamphiphilic compound, which combines the molecular structure of the drugs zidovudine and didanosine as polar subunits and a phosphoryl-deoxycholyl (**5**) as a nonpolar subunit, was synthesized and evaluated against HIV-1 by Jin et al. (2011) (Figure 1) [22]. The prodrug (**5**) formed small spherical vesicles with an average size of 174 nm and high zeta potential (−31.3 mV), which caused repulsion between particles. The degradation of (**5**) was highly pH-dependent. It degraded at pH 2.0 (representing the stomach environment) in 0.5 h. However, it was stable in a neutral medium with a degradation half-life (t_1/2_) of 61.3 h at pH 5.5 (intestine) and t_1/2_ of >1000 h at pH 7.4 (blood). Studies on the stability of (**5**) in the plasma of mice, dogs, monkeys, and humans showed that it was highly stable under most conditions; however, in mouse plasma, its t_1/2_ was only 9 h. The anti-HIV effect of (**5**) is similar to that of AZT; the EC_50_ of (**5**) is 5 nM in infected MT4 cells. The half toxic concentration (TC_50_) of (**5**) was found to be 42.22 µM (SI: 8444). Pharmacokinetic studies showed that the bioavailability of (**5**) was 30.8% and 90.5% after oral and intraperitoneal administration, respectively [22]. The low oral bioavailability could be explained by certain chemical and enzymatic instability in this acidic environment [22]. High levels of (**5**) were found in organs such as the lungs, liver, thymus, and lymph nodes [22]. The degradation of (**5**) releases the parental drug AZT quickly and the drug didanosine (ddI) slowly. In mice, (**5**) administered via the oral route showed a t_1/2_ of 15.7 min, C_max_ of 10 µg/mL, and area under the curve (AUC)0–120/µg·mL^−1^/min of 575; when (**5**) was administered intraperitoneally, those values were 28.1 min, 41.4 µg/mL, and 1690 µg·mL^−1^/min, respectively [22]. This prodrug can simultaneously deliver two different drugs to the target cells; thus, it can enhance the efficacy and concentration of those drugs in the target organs [22].

Nikavir (5′-*H*-phosphonate) is a prodrug of AZT that was approved in Russia in 1999 (Figure 1). The insertion of a phosphate group into the 5′ position of AZT reduced the cytotoxicity in comparison to AZT. This, in turn, decreased blood count shifts and improved tolerance in patients during long-term treatments. Nikavir is four times less toxic than AZT, although it is one order of magnitude less efficacious. In vivo, nikavir is converted to AZT-5′-triphosphate, which inhibits viral replication. The decrease in cytotoxic effects is attributed to the pharmacokinetic parameters; the drug is slowly converted to AZT, which prevents high levels of the drug from accumulating in plasma [23,24]. Analogs of nikavir were prepared and evaluated to comprehend the structure–activity relationship (SAR) by El-Sayed et al. (2015). These analogs showed low activity and low toxicity. Derivative (**6**) (containing the Cl_3_C group) (Figure 1) had the best profile with an infective dose (ID_50_) of 0.45 µM, a CC_50_ of 1.34 µM, and an SI of 3000 [25].

NeuroAIDS, also known as HIV-associated neurocognitive disorder (HAND), has a complex and wide range of clinical symptoms characterized by cognitive and neurological issues commonly found in HIV-1-infected patients in the advanced stage. NeuroAIDS causes symptoms that include anxiety, depression, loss of attention, difficulty in memorizing, motor coordination, dementia, etc., which decreases the quality of life of patients [26,27]. Cognitive impairment develops in up to 50% of HIV-positive patients, even after initiating ART [28]. Chronic inflammation induced by HIV-1 can lead to blood–brain barrier damage, thus facilitating the entry of harmful substances into the central nervous system (CNS). The exact mechanism underlying neuroAIDS is not clear, but it is known that antiretroviral therapy can decrease the rate of progression of the infection and control inflammation. 

Many studies have suggested the potential harmful effects of antiretroviral therapy on the brain [29]. This finding reinforces that a thorough assessment of the penetration of drugs into the CNS and the effects of these drugs is necessary. Some studies have proposed strategies to improve the delivery of antiretroviral drugs to the CNS [30].

Mdanda et al. (2019) carried out studies on rats and showed that a single dose of AZT (50 mg/Kg) could achieve a C_max_ value in plasma of up to 55,976 ng/mL, whereas, for brain tissue, this concentration was 692 ng/mL [31]. To improve access to the CNS, AZT and ursodeoxycholic acid (UDCA) were conjugated by Dalpiaz et al. (2012), to circumvent the active efflux transport system (AET) of the central nervous system (CNS). This specific conjugation occurred at the 5′-ester of AZT with UDCA, which has antiapoptotic effects. The results of the high-performance liquid chromatography (HPLC) analysis revealed that the half-life (t_1/2_) of the prodrug in rat blood was approximately 10 s, while in human plasma, it had a longer half-life of 7.53 h; thus, it exhibited controlled release of AZT. Upon administration, the prodrug (UDCA-AZT) was detected in the rat brain, where it underwent rapid hydrolysis and showed a t_1/2_ of 7.24 min. In liver homogenates, it had a t_1/2_ of 2.70 min, suggesting that it might be released in this compartment. The permeation characteristics of the compound were investigated across physiological barriers using a human retinal pigment epithelium (HRPE) cell line. A polarized cell monolayer was formed with epithelial features. Compared to the active efflux of AZT from the CNS, the bidirectional permeation of 30 M AZT across this cell monolayer was governed by apparent permeability coefficients (PEs) that were higher from the apical to the basolateral compartments (PE = 209 ± 4 × 10^−5^ cm/min) than in the reverse direction (PE = 133 ± 8 × 10^−5^ cm/min). In contrast, the permeability coefficients of 30 M UDCA-AZT were similar for inflow (PE = 39.1 × 10^−5^ cm/min) and efflux (PE = 31.3 × 10^−5^ cm/min), which suggested that the prodrug could accumulate in the CNS. Partial hydrolysis of the prodrug during its penetration through the cell membrane was associated with its relatively lower PE values. The molecular structure of UDCA might serve as a reference for designing drugs or prodrugs to evade active efflux transport systems, thus minimizing the chances of developing multidrug resistance. Neither UDCA nor UDCA-AZT showed interactions with the active efflux transport system [32].

To improve the effect of the prodrug UDCA-AZT, the strategy of using a delivery system with solid lipid nanoparticles for nasal administration was developed. Nasal administration is an effective strategy to increase permeation and brain uptake [33]. Medicines that can adhere to the olfactory mucosa can traverse into the cerebrospinal fluid (CSF) by diffusing across the mucosal layer. Additionally, the drug can enter the brain through the trigeminal or olfactory nerves that extend to the nasal cavity. These nanoparticles were spherical, and their diameters were approximately 7–16 µm. Dalpiaz et al. (2014) found that for the UDCA-AZT prodrug, the loading rate was 0.57% (*w*/*w*) when tristearin was used and 1.84% (*w*/*w*) when stearic acid was used [34]. The main difference between these two methods is that tristearin can control the rate of release, whereas stearic acid can improve the dissolution rate of the system [34]. The solid lipid nanoparticles containing tristearin demonstrated a significant enhancement in prodrug stability with 75% retention after 30 min, whereas, in the presence of stearic acid, only 14% retention was recorded after 30 min [34]. The nasal administration of stearic acid-based solid lipid nanoparticles facilitated prodrug uptake into the CSF, revealing a direct pathway from the nose to the CNS. The uptake of the prodrug improved after chitosan was included. By implementing this method, the uptake of the prodrug in CSF increased sixfold (up to 1.5 µg/mL within 150 min) after the postnasal administration (at 200 µg) [34].

### 2.2. Tenofovir Prodrugs

In 1993, some researchers reported different effects of enantiomers of acyclic nucleoside phosphonate—tenofovir (TFV)—against herpes virus and retrovirus. Although the drug has promising in vitro effects, at physiological pH 7.4, it is a negatively charged dianion due to the presence of a phosphate group. This property decreases membrane permeability and, consequently, reduces oral bioavailability. Efforts to improve such limitations led to the synthesis of tenofovir disoproxil (TFV-DP), a tenofovir prodrug with higher oral bioavailability (~25%) than the parental drug tenofovir. The presence of food can increase the bioavailability by up to 35%. After oral administration, TFV-DP is hydrolyzed by plasma esterases and then phosphorylated by cellular kinases to the active metabolite tenofovir diphosphate (Figure 2). TFV-DP is commonly synthesized as its fumarate salt and has favorable pharmacokinetic properties; it is generally administered at a dose of one tablet a day. The drug, used in combination, is effective in treating naïve patients. It is administered in most first-line treatments. However, its long-term administration is associated with kidney injury and a decrease in bone mineral density. Disturbances in the tenofovir secretory pathway (higher OAT-1 activity or lower MRP efflux transport activity) can increase the concentrations of TFV in the cell, which can cause degradation and dysfunction of mitochondrial DNA and proximal tubulopathy. This is a serious concern for the long-term treatment of HIV, and strategies to reduce this inconvenient effect might be investigated using the prodrug approach [35].

Tenofovir alafenamide (TAF), approved by the FDA in 2016, is a prodrug of tenofovir that contains a phenoxy isopropylalanineamidate (alafenamide) subunit (Figure 3). In vitro studies have shown that TAF is approximately 10-fold more active than TFV-DP and up to 1000-fold more active than parental TFV. ProTide, which exhibits a promising application to allow the delivery of monophosphates and monophosphate nucleoside intracellularly, improving the tissue exposure and oral bioavailability of drugs and reducing the collateral effects on nontarget tissue [36]. TAF efficiently delivers the nucleoside inside cells at high concentrations, which decreases the plasma levels of TFV and the adverse effects on the kidneys and bones. TAF can accumulate in some compartments, such as the lymphatic tissue and the liver, which is highly desirable for HIV. Studies on the evaluation of the permeability and stability of TAF have shown that ProTide uses systemic circulation to effectively load target cells by saturating intestinal efflux transporters; this process is facilitated by its high water-solubility. Babusis et al. (2013) conducted a study in which TAF exhibited concentration-dependent permeability across the monolayers of Caco-2 cells, as well as oral bioavailability ranging from 1.7% at 2 mg/kg to 24.7% at 20 mg/kg. In intestinal and hepatic extraction in the portal vein of cannulated dogs, its half-life ranged from 0.12 h at 0.5 mg/kg (iv) to 0.34 h at 20 mg/kg (po). The systemic pharmacokinetic profile of TAF in dogs showed that after oral administration of the drug at 5 mg/kg, it was rapidly absorbed. The highest plasma concentration of the drug recorded was 1.58 µM 0.14 h after administration. TAF disappeared quickly (2 h), and this change was associated with an increase in the concentration of TFV in plasma. Exposure of TAF to plasma led to the rapid accumulation of TFV-DP in PBMCs (18 µM after 1 h) [37]. 

In clinical trials, TAF was well tolerated and presented potent antiviral efficacy. Approximately 8 mg of TAF had antiviral effects similar to those of 300 mg of TFV-DP. Administering **TAF** increased intracellular levels of TFV-DP in peripheral blood mononuclear cells (PMBCs), which confirmed the observations of other in vitro studies [38,39,40,41,42]. 

A comparative study on the tenofovir prodrugs, TAF and TDF, was conducted by Callebout et al. (2015). In that study, TAF showed better results than TDF, with higher antiviral activity against HIV-1 and HIV-2; the EC_50_ ranged from 0.10 to 12.0 nM (mean EC_50_ was 3.5 nM). For HIV-2 isolates, the mean EC_50_ was 1.8 nM for TAF and 6.4 nM for AZT (used as a control group) [42]. A year prior, Markowitz et al. (2014) conducted other clinical studies that demonstrated comparable findings. The researchers also found mild adverse effects (headache, nausea, and flatulence) after administering TAF but not TFV-DP. TAF was also more effective in reducing the viral load (HIV-1 RNAm) and did not result in resistance mutations [39].

ProTide can be prepared using three distinct routes. First, the coupling reaction of diarylphosphite with a nucleoside followed by oxidation of the amino group; second, the treatment of phosphorochloridate with the desired nucleoside; third, the treatment of arylphosphate of a certain nucleoside with an amino acid through a coupling reaction [36]. By masking the monophosphate group, which is negatively charged at a physiological pH of 7.4, the poor cellular uptake of nucleosides can be decreased. The activation of ProTide inside the cell involves the cleavage of the ester by esterases. Two main esterases are involved in cleaving esters and include carboxylesterase 1 (CES1) and cathepsin A (Figure 3). The nature of the ester group can influence the lipophilicity, along with the steric and electronic parameters, leading to the release of carboxy metabolites at different rates. In the second step, the negative charge of the amino acid facilitates a nucleophilic attack on the phosphonate or phosphate that contains a partial positive charge at this electron-deficient site. This results in the elimination of the aryl-leaving group and subsequent cyclization (Figure 3). This step leads to the formation of an unstable five-membered ring that opens after hydrolysis and forms a phosphoramide intermediate (Figure 3). In the last step, a phosphoramide-type enzyme once again hydrolyzes the phosphorus–nitrogen (P-N) bond, producing the triphosphorylated nucleoside (Figure 3). This nucleoside is then released by cellular kinases, enabling its effectiveness against virus-encoded DNA or RNA polymerases (Figure 3) [43,44]. 

The tyrosine-based prodrug (*ProTide* strategy) of TAF, in which the phenol group of the traditional compound was replaced by a tyrosine derivative, was identified as a ketone-containing compound (**7**), which showed high antiviral potency against HIV-1. The study developed by Duwal et al. (2012) with prodrug (**7**) (Figure 4) (slow-eluting epimer) showed an EC_50_ value of 6 pM against HIV-1 and cytotoxicity with a CC_50_ of 9.2 µM. The SI of this compound was 1500.000, which was an approximately 300-fold increase in SI compared to TAF (EC_50_ of 0.0064 µM; cytotoxicity with CC_50_ of 35 µM). In a metabolomics study in which lymphoid cells (CCRF-CEM) were used, (**7**) showed higher cellular uptake. The intracellular concentration of the prodrug was 102.36 pmol/10^6^ cells, with a higher concentration of TDF (148.09 pmol/10^6^ cells) than the parent drug TAF. Thus, the ProTide (**7**) showed higher antiviral activity. These findings highlighted that (**7**) is an effective pro-pharmaceutical agent that can release a sufficient amount of TDF [41].

Cobb et al. (2021) modified the ProTide technology to form stable nanocrystals. Phenylalanine and alanine amino esters containing long-chain fatty acids were synthesized to investigate the ProTide approach. Both prodrugs, i.e., (**8**) and (**9**) (Figure 4), showed higher lipophilicity than tenofovir. Both compounds also showed higher intracellular drug uptake and retention, mainly in the important compartments for HIV, such as the liver, spleen, lymph nodes, gut, lymphocytes, and rectal and vaginal tissues. In vivo experiments using Sprague Dawley rats showed that after intramuscular injection of both prodrugs, a sustainable level of tenofovir-diphosphate was maintained at four times the effective dose over two months of the study. These findings were promising, including for antiviral preexposure prophylaxis (PrEP), considering that TDF is the choice for PrEP therapy in many countries [45]. 

Prodrugs containing drug-lipid conjugates can effectively enhance lipophilicity, enable sustained release of the parental drug, elevate concentrations in specific body compartments (organs, tissues, and cells) through targeted drug delivery, and offer the potential to alter the route of administration. These prodrugs are being extensively investigated in the field of drug discovery [46]. Drug–lipid conjugation can be performed using different lipophilic substances, such as fatty acids (e.g., squalene, stearic acid, oleic acid, palmitic acid, linoleic acid, lauric acid, and octadecanoic acid), steroids (cholesterol, ursodeoxycholic acid, and lithocholic acid), glycerides, and phospholipids. The choice of the lipid-acid chain depends on the physicochemical properties, stability, and additional biological effects of the designed prodrug. Additionally, distinct chemical bonds might be assessed to design conjugates consisting of esters, amides, hydrazone, disulfide, carbamates, carbonates, and phosphodiesters. The selection of this chemical bond is determined by considerations of reactivity, stability, selective delivery, drug release rates, and modulation of pharmacokinetic properties [46]. Among the bonds, esters and amides are the most studied. Amide bonds are more stable and exhibit slower rates of hydrolysis than ester bonds. Other examples include the hydrazone bond, in which the sensitivity to acidic environments, such as that found in lysosomes, can lead to the hydrolysis and release of the drug and the lipid [46]. 

The lymphatic system is an important site for HIV viral replication. Some studies have found a correlation between the concentration of antiviral drugs in lymphoid tissues and the suppression of HIV, as well as viral persistence [47,48].

Many researchers have used lipid conjugates and the prodrug approach to deliver drugs to the HIV reservoir in the gut lymphatics using dendrimers, nanoparticles, nanosuspensions, micelles, conjugated polymers, etc. [49]. During lipid metabolism, lipophilic drugs can be organized and packaged into intestinal lipoproteins as chylomicrons and secreted into the mesenteric lymph (Figure 5) [49]. These drugs can enter the systemic circulation through the thoracic lymphatic duct (Figure 5) [49]. Several studies have shown that highly lipophilic compounds with suitable physicochemical characteristics (log D7.4 > 5, Log *p* > 5, large particles, solubility > 50 mg/mL) can be effectively transported by the lymphatic system through the intestinal tract when administered orally with a high-fat diet (Figure 5) [49]. Antiretrovirals, for example, when absorbed by the intestinal lymphatic vessels, are organized in lipid vesicles in the enterocyte and transported by lymph to the lymph nodes (Figure 5). Thus, they are distributed according to the lymph flow pattern (Figure 5). Lipophilic antiretrovirals are more likely to perfuse through the middle of the lymph node, where cell density is high and fluid flow is low [50].

This type of lymphatic transport can avoid a first-pass effect. The reaching of the lymphatic system is a result of triglycerides being re-esterified after their hydrolysis in the small intestine by pancreatic lipases. This results in the formation of metabolites (2-monoglycerides and fatty acids) that are absorbed by enterocytes and re-esterified into triglycerides, which are organized into chylomicrons and secreted into the mesenteric lymph nodes. Lymph nodes are tissues where viral RNA and DNA are found in nonhuman primates, humanized mice, and humans undergoing antiretroviral therapy. Thus, lymph nodes are important in the pathology of HIV, as they contain large populations of CD4+ T cells and other immune cells organized in specific regions that facilitate the immune response. HIV spreads to regional lymph nodes within 3–6 days after the infection starts, due to its large target population of CD4+ T cells and the constant movement of cells by elevated endothelial venules (HEVs) and lymphatic vessels. Systemic spread of HIV occurs within 6–25 days. This reservoir can be maintained by two mechanisms during antiretroviral therapy, including low levels of viral replication and clonal expansion of infected latent CD4+ T cells. The incomplete inhibition of viral replication in lymph nodes might be due to the low concentration of antiretroviral agents in this tissue compared to their concentration in plasma [51,52]

TFV lipid conjugates were specifically designed and synthesized by Lanier et al. (2010) to mimic lysophosphatidylcholine, aiming to target the natural lipid pathway and attain elevated intracellular concentrations of the active antiviral agent [53]. This approach enhances the effectiveness against both wild-type and mutant HIV strains.

TDF-exalidex (**10**) (Figure 6), {3-(hexadecyloxy)propyl hydrogen [(*R*)-1-(6-amino-9*H*-purine 9-yl)propan-2-yloxy]methylphosphonate. Hexadecyloxypropyl TFV [HDP-TFV] was found to be active against all major HIV-1 subtypes in PBMCs, with EC_50_ values ranging from 0.20 to 7.2 nM for the 27 different strains of HIV evaluated. The compound (**10**) is a lysoglycerophospholipid conjugate prodrug designed to increase intracellular concentrations of TFV and decrease the circulating levels and systemic presence of TFV. Compound (**10**) has low bone and renal toxicity that is often reported following the long-term administration of TFV. In vitro studies have also shown that this lipid prodrug is 97-fold more active against HBV and up to 267-fold more active against HIV-1 [53,54]. This prodrug mimics the uptake pathways of lysolecithin in the gut. Compound (**10**) can be cleaved by phospholipase C to release the parental drug TFV; however, the long-chain fatty acid is metabolized by CYP450 omega-hydrolylases in the liver. These undesired reactions hinder access to HIV-1-infected cells, and the resulting metabolite may have potential organ toxicity risks [53,55]. Pharmacokinetic studies revealed substantial hepatic extraction, implying that this approach may find more promising applications in the treatment of hepatitis rather than HIV infection [56].

These results suggested that the cellular uptake of compound (**10**) was high, which resulted in higher intracellular levels of the active antiviral anabolic agent, TFV-diphosphate (TFV-PP). No cytotoxic effects were observed up to the high-tested concentration of 1000 nM. It also showed activity against HIV-2 in PBMCs (mean EC_50_ ± SD = 3.5 ± 1.5) and against six HIV-1 isolates in MDMs (mean EC_50_ ± SD = 2.5 ± 1.7 nM) [57]. Human PBMCs were used to evaluate the relative levels of TFV-PP produced by (**10**) and TFV. Human PBMCs exposed to 1 µM (**10**) for 24 h produced approximately 34 times more TFV-PP than that produced after exposure to 1 µM of TFV. The lipid–TFV conjugate is not a substrate for human organic anion transporters (hOATs 1 to 3). After absorption, it is activated in human PBMCs, which increases its efficacy and decreases the toxicity imparted by TFV. No antagonistic interactions were recorded between (**10**) and any FDA-approved antiretroviral drug, such as NRTI. Thus, the prodrug could deliver high levels of the active metabolite and reduce toxicity [57].

To avoid the metabolic issues related to (**10**), Pribut et al. (2021) synthesized and evaluated ω-functionalized lysoglycerophospholipid prodrugs derived from TDF-exalidex to improve hepatic stability, decrease TFV organ toxicity, and reduce CYP-mediated ω-hydroxylation. Optimizing the length of the chain was expected to improve hepatic stability and increase the half-life. From the series prepared, two compounds (**11** and **12**) (Figure 6) showed the best pharmacokinetic profiles. The saturated CF_3_ derivative (**11**) exhibited an IC_50_ value of 0.049 µM for HIV-1 and a CC_50_ value > 100 µM using human embryonic kidney (HEK293T) cells. For this prodrug (**11**), the therapeutic index (CC_50_/HIV IC_50_) was found to be greater than 2040. Prodrug (**11**) exhibited metabolic stability using mouse liver microsome experiments with a half-life (t1/2) greater than 120 min. The values for the inhibition of CYP 2D6 and 3A4 were greater than 20 µM [58]. The TMS acetylenyl prodrug (**12**) showed an IC_50_ value of 0.069 µM and a CC_50_ value >100 µM (HEK293T cells), with a therapeutic index greater than 1450. The T_1/2_ was higher than 120 min, and the IC_50_ values for CYP 2D6 and 3A4 were 21.7 µM and >20 µM, respectively. Prodrug (**12**) presented a higher metabolic stability than TDF-exalidex, which in the same experiments showed an IC_50_ value of 0.018 µM for HIV-1, a CC_50_ value of 97.6 ± 2.68 (therapeutic index of 5420), and t1/2 of 42 min. Preliminary SAR revealed that chain lengths of 18 and 20 atoms were optimal for antiviral activity. Analogs containing ether or thioether linkers had equal metabolic stability but higher antiviral activity than methylene linkers. The ω-terminal functionalized groups, such as acetylenyl, TMS acetylenyl, CF_3_, and CF_3_ acetylenyl, showed higher metabolic stability in HLM compared to prodrugs with unfunctionalized terminals, probably due to a decrease in the metabolism by ω-oxidases. The antiviral activity of the substituents TMS acetylenyl, CF_3_, and CF_3_ acetylenyl derivatives was similar to that of TDF-exalidex, whereas undecorated acetylenyl congeners showed significantly less antiviral activity. Such differences occurred due to the variations in cellular permeability and metabolism. The mouse plasma pharmacokinetic profile showed an AUC0−24 h (h·ng/mL) of 301, 1280, and 550 for the compounds TDF-exalidex, prodrug (**11**), and prodrug (**12**), respectively. Such effects suggested a decrease in the metabolism catalyzed by ω-oxidases [58].

Giesler et al. (2016) examined the effect of modifying the length of the carbon chain by performing bioisosteric replacement of the carbon–carbon bond to a disulfide bond [56]. Lipid phosphomonoester disulfide-linked prodrugs do not rely solely on reduction for prodrug cleavage; instead, enzymatic activation is required to release TFV into the target cell. The researchers suggest that these prodrugs likely depend on enzymatic hydrolysis to facilitate S-S bond reduction. The researchers prepared two different series. In the first study, they found that the prodrug (**13**) ((*R*)-4-(hexadecyldisulfanyl)butyl (((1-(6-amino-9*H*-purin-9-yl)-propan-2-yl)oxy)methyl)phosphonate) (Figure 6) exhibited anti-HIV effects with an EC_50_ value less than 0.5 nM, a CC_50_ value greater than 50 µM, and a therapeutic index greater than 100,000 [50]. Prodrug (**13**) was more efficacious than TFV (EC_50_ = 0.320 µM, CC_50_ > 100 µM, and TI > 300). Prodrug (**13**) was activated by 5-exotet cyclization to yield nonelectrophilic tetrahydrothiophene after enzymatic hydrolysis, in which phospholipase C and/or sphingomyelinase may have participated [56]. The researchers hypothesized that its high antiviral activity was due to the presence of a “twisted” S^_^S bond in this prodrug. The bond involved a 90° dihedral angle that increased membrane fluidity to accelerate the translocation of the conjugate from the outer side of the plasma membrane to the inner side, where it was enzymatically cleaved to release TFV into the cytosol [50]. However, this disulfide conjugate showed low aqueous solubility, and the mutagenic potential of thiirane formed by the decomposition of *S*-acyl-2-thioethyl (SATE) and dithioethanol (DTE) encouraged further investigation of ways to mitigate this deleterious effect [56].

In the second series, Giesler et al. (2016) developed lipid phosphomonoester disulfide-linked prodrugs that did not require enzymatic activation to release TFV into the target cell [59]. Lipid conjugates (**14** and **16**) (Figure 6) had alkyl chains of 14 to 18 carbon atoms and showed maximal antiviral activity against HIV, with EC_50_ values in PBMCs of 0.5 nM and 0.6 nM, respectively. Compounds (**14**) and (**15**) displayed robust stability in human plasma with half-lives of >24 h, probably due to the formation of micelles that protected the fragile S-S bond from the action of reductases circulating in human serum [59]. Stability assays showed that (**14**) and (**15**) exhibited similar stabilities with t_1/2_ > 2 h. After optimization, the lipid conjugates significantly improved the potency of the original nucleoside by more than 600-fold, showing that the compounds had adequate stability [59]. 

### 2.3. Lamivudine Prodrugs

Lamivudine, also known as 2′,3′-dideoxy-3′-thiacytidine (3TC), was designed by the bioisosteric replacement of the 3′-methylene hydroxide group with a sulfur atom in the ribose scaffold [60]. The drug is efficaciously absorbed after oral administration, and its bioavailability is 86%. Nevertheless, the prodrug approach must address challenges such as peripheral neuropathy and gastrointestinal issues (e.g., nausea, diarrhea, and vomiting) to ensure the sustained, long-term use of **3TC**.

Khandazhinskaya et al. (2011) synthesized phosphonate-based derivatives of 3TC, and their anti-HIV activity was assessed using infected MT-4 cells. The IC_50_ value for the most active prodrug (i.e., compound (**17**)) (Figure 7) was 3.4 µM, which was considerably higher than that of 3TC (IC_50_ = 0.44 µM). However, compound (**17**) exhibited lower toxicity (CC_50_ > 3410 µM) than 3TC (EC_50_ = 43.6 µM) and had a selective index (SI) one order of magnitude greater than that of 3TC. The stability of these prodrug compounds was evaluated across a pH range of 5.5–8.5; the half-life was found to be slightly over 24 h. In the whole blood of dogs, compound (**17**) showed a half-life of 5 h. The pharmacokinetic profiles of compound (**17**) were similar when the compound was administered orally to dogs and intragastrically to rabbits. Following administration, the t_max_ values increased 2–3 times, C_max_ was nearly halved, and the AUC was reduced by 1.5–2 times compared to 3TC administered alone [61].

The lipophilicity attributed to the length of the carbon chain at the 5′-*O*-position was modulated for lamivudine derivatives to improve their capacity to cross the cellular membrane through passive diffusion, and they were further studied to design new prodrugs. Thus, Ravetti et al. (2009) prepared a series of 3TC carbonate prodrugs using the coupling agent *N*,*N*-carbonyldiimidazole and certain alcohols [62]. The effectiveness of these prodrugs was evaluated against HIV-1-infected cells. Compared to 3TC with a log *p* value of −0.91, the carbonate prodrugs were more lipophilic and showed log *p* values ranging from −0.81 to 1.72 [62]. Some derivatives had IC_50_ values higher than that of 3TC; for example, the prodrug (compound **18**) (Figure 7) showed an IC_50_ of 0.065 µM and SI greater than 19,700. In vitro assessments in PBMCs highlighted the relationship between the increase in cytotoxicity and the elongation of the carbon chain of the alcohol moiety, suggesting that a chain containing five carbon atoms is ideal for such effects [63]. 

After lamivudine was used against HIV, its efficacy against the hepatitis B virus (HBV) was assessed. Studies have shown that 3TC is effective in reducing HBV replication. To improve the lipophilicity of the drug, Li et al. (2010) designed 3TC prodrugs (compounds **19** and **20**) in which stearate was used as the carrier. This approach increased the log *p* value from −0.95 3TC to 1.82 (prodrug). Then, the researchers developed a particulate delivery system to improve the uptake of the prodrug using stearic acid-g-chitosan oligosaccharide (CSO-SA) micelles. After the stearic-3TC prodrug was incorporated into micelles (average size: 460 nm), the zeta potential increased (29.7 mV). This modification increased its uptake by HepG2.2.15, decreased its cytotoxicity, and exhibited a linear kinetic of hydrolysis to release 3TC; these changes helped in reducing the viral infection [63].

A liver-specific prodrug (**21**) (Figure 7) derived from the antiviral compound lamivudine 3TC was prepared through the coupling of 3TC to dextran (~25 kDa) using succinate as the spacer. The synthesized (**21**) had >99% purity and a substitution degree of 6.5 mg of **3TC** per 100 mg of the conjugate. Dextrans are promising systems/carriers that might be further investigated in the field of drug discovery, as plasma kinetics and distribution of dextran carriers in tissues depend on the molecular weight (MW) of the polymer. For example, dextrans with an MW of 20–70 kDa showed high specificity for the liver, probably because of their size, which permitted their passage through the larger pores of the liver sinusoids [64]. The prodrug showed high chemical stability under different pH conditions (4.4 and 7.4) and in rat blood, releasing 3TC at slow rates. Plasma concentration-time curves after IV administration were also examined, and the results showed that for 3TC, C_max_ was 4.94 µg·mL^−1^ and t_1/2_ was 37.9 min, whereas, for the prodrug, C_max_ was 52.5 µg·mL^−1^ and t_1/2_ was 125 min, respectively. The linkage of 3TC to dextran considerably decreased drug clearance and distribution volume by factors of 40 and 7, respectively. The prodrug level was 50 times higher than the level of the parental drug 3TC in the liver. The prodrug concentration was also high in the kidney, but it was almost absent in the lungs, heart, brain, and spleen [65].

Drug-releasing systems, such as hydrogels, nanogels, and polymer conjugates, are effective ways to prolong the half-life and reduce the cytotoxic effects of nucleosides. In certain scenarios, the kinetics of hydrolysis are contingent upon the chemical nature and stability of the bond between the drug and the system, which plays a pivotal role in regulating drug delivery. Polymers containing hydroxyl and sulfonated side chains linked to 3TC through a disulfide self-immolative linker were synthesized, characterized, and evaluated against HIV-1 infected cells (HeLa-derived TZM-bl cells). Polymer sulfonation is important to avoid HIV-1 entry; approximately 50% of sulfonated monomers were found to inhibit reverse transcriptase. Sulfonate monomers also exhibit activity against DNA−DNA polymerase, which further enhances its antiviral effects. These mechanisms together considerably improve the antiviral effects. Thus, incorporating such mechanisms while developing prodrugs can greatly enhance their half-life and decrease their cytotoxicity [66]. 

Drug delivery to specific compartments, such as the liver, is a promising approach to enhance the effectiveness of treatment. This strategy ensures higher concentrations of the active antiviral agent in the liver with minimum exposure of other tissues to the drug. HepDirect, a liver-targeting approach, is a prodrug strategy developed to achieve these aims. In this approach, the chemical structure of the antiviral drug is modified to develop a compound that remains inactive until it reaches its target, i.e., the liver. In the liver, the prodrug undergoes a specific biochemical transformation triggered by liver enzymes, mainly CYP450 isoforms [67,68]. This transformation converts the prodrug into its active form, which leads to the release of the antiviral agent at the precise site of infection [67,68]. Studies have shown that this technique has higher antiviral activity and a better safety profile than the traditional systemic administration of antiviral agents [67,68].

Gualdesi et al. (2014) synthesized 3TC prodrugs containing 5′-O-carbonates to perform preformulation studies [69]. The water solubility of all prodrugs ranged from 0.24 to 17 mg/mL, whereas for 3TC, the solubility was 65.9 mg/mL at 25 °C. The increase in the carbon chain length of the substituent led to a reduction in aqueous solubility. Most of the 5′O-carbonate prodrugs showed favorable chemical stability, indicating that their structures affected the hydrolysis rate. At pH 7.4, the half-life for all prodrugs ranged from 22.85 h to 378.77 h, whereas, for enzymatic hydrolysis, the half-life ranged from 5.44 min to 35.20 min [69]. The log *p* values for prodrugs were found to vary between −0.81 and 1.72 [69]. The intestinal permeability of two of these prodrugs (**22** and **23**) (Figure 7) was evaluated, and the results showed a twofold and tenfold increase, respectively, compared to the intestinal permeability of 3TC **[69]**. However, (**22**) was a substrate of P-glycoprotein, and (**23**) underwent enzymatic hydrolysis during permeation [62]. Even small chemical modifications, using a homology series, for example, can greatly affect the physicochemical properties and the kinetics of hydrolysis. Thus, chemical modification is an important parameter to evaluate the design of new prodrugs [70].

The efficacy of 3TC therapy for HBV is not very high, and the virus shows considerable drug resistance. This is probably because of the poor bioconversion of 3TC to the triphosphate form that is necessary to inhibit HBV DNA polymerase. As the HepDirect prodrug is resistant to esterase cleavage, it first needs to undergo oxidative cleavage by CYP450 (CYP3A4) to be converted into nucleoside monophosphate and then into the active nucleoside triphosphate. Reddy et al. (2005) found that the 3TC HepDirect prodrug (compound (**24**)) can increase the concentration of (3TC)-triphosphate by up to 35-fold in rat hepatocytes. This prodrug also showed a sustainable release in rat hepatocytes compared to 3TC. After IV administration in rats, the concentration of the prodrug in the liver was 29.4 nmol·g/h, and that of **3TC** was 2.6 nmol·g/h. The concentration of 3TC after prodrug metabolism in plasma was <12.8 µM, indicating that its concentration in the liver was >320-fold higher than the concentration of 3TC [71]. 

Codrugs, also known as mutual prodrugs, have two or more drugs/active compounds covalently bound through a spacer link or a direct link. The primary goal of codrug development is to achieve synergistic effects by combining compounds that target different aspects of a disease or its underlying mechanisms. This approach can improve therapeutic outcomes, decrease drug resistance, and overcome the limitations associated with single-drug therapy. Codrugs can be especially effective in treating complex diseases, where multiple pathways contribute to the pathogenesis, or where single-drug treatments have limited efficacy. However, finding the balance between the chemical properties of the individual drugs is a major challenge in the development of codrugs. Thus, the stability, solubility, and pharmacokinetics of both active compounds should be compatible [72,73].

Rossi et al. (2007) developed mutual prodrugs of 3TC and TDF to obtain a heterodinucleotide (**25**) (Figure 7) [74]. The ability of this prodrug to protect against ‘de novo’ HIV-1 infection was evaluated. Cells were incubated with parental drugs, the prodrug, and the combination (**3TC** + tenofovir) for 18 h before HIV-1 infection, and then, viral production was evaluated for 35 days. Although the combination decreased viral production by ~40.7%, the prodrug was more active, leading to only 8.5% viral production in 35 days. The activation of (**25**) was evaluated using murine plasma, which contains enzymes capable of cleaving the phosphate bridge of the prodrug. The results showed that (**25**) was stable in murine plasma (t_1/2_ = 14 h), and its half-life was higher than that of (3TC) [74]. The prodrug was stable in erythrocytes. Experiments in which prodrug-loaded erythrocytes were added to macrophages showed that the treatment had a protective effect against viral infection [74]. However, pharmacokinetic investigations in mice showed that (**25**) was quickly removed from the bloodstream (t_1/2_ = 15 min), and it did not have any benefits compared to the administration of individual drugs [74].

Another codrug, involving 3TC and ursolic acid, was synthesized and evaluated. Ursolic acid has hepatoprotective effects. Zhong et al. (2012) hypothesized that the codrug might be effective against infectious diseases such as hepatitis. Its chemical stability was studied at pH 7.4, using plasma and in the presence of lipase. The researchers found that (**26**) (Figure 7) was rapidly hydrolyzed in the presence of lipase, exhibiting a half-life of 1.4 h, which was shorter than its half-life in phosphate buffer (pH 7.4) (half-life of 11.2 h) and in buffered human plasma (half-life of 5.4 h). Studies on its chemical stability at pH 3–6 showed that the codrug was highly stable, and its half-life was greater than 40 h. Hepatoprotective effects were determined using the acute liver injury model induced chemically by carbon tetrachloride (CCl_4_) and acetaminophen. Animals treated orally with 200 µmol/Kg of the codrug showed effects similar to those found for the reference hepatoprotective compound ursolic acid. Histopathologic studies showed a decrease in inflammation and hepatocellular necrosis in the codrug group. Anti-HBV effects were investigated in vivo using the DHBV-infected duckling model. Animals treated with the codrug at 150 and 250 mg/kg/d showed lower levels of DHBV-DNA, as determined by PCR. These findings suggested that the 3TC–ursolic acid codrug should be investigated for HBV infections [75].

### 2.4. Stavudine (d4T) Prodrugs

Stavudine (2′,3′-didehydro-2′,3′-dideoxythymidine, d4T) is a pyrimidine nucleoside, which differs from thymidine in that the 3′-hydroxyl group is replaced by a hydrogen atom and a double bond is present in the ribose ring between the 2′ and 3′ positions. The bioavailability is estimated to be approximately 85% after oral administration; however, the drug has a short half-life (~1.5 h) [69]. In contrast, the triphosphate metabolite has a half-life of 3.5 h [76]. 

Myristoylated prodrugs derived from d4T (stavudine) were designed and synthesized by Singh et al. (2014). Among these prodrugs, the bifunctional compound (**27**) (Figure 8) containing a C10 alkyl chain exhibited the most potent antiviral activity against HIV-1. In MT-4 cells, it showed a very low EC_50_ of 0.03 µM and minimal cytotoxicity. Its CC_50_ exceeded 16.85 µM in MT-4 cells. These properties were more favorable than those found for stavudine. This compound could inhibit HIV replication at two different stages: reverse transcription and posttranscriptional processing of various proteins. In contrast, stavudine displayed an EC_50_ of 0.15 µM and some level of cytotoxicity, indicated by a CC_50_ of 41.60 µM, in MT-4 cells [77].

Rodríguez-Pérez et al. (2010) synthesized a series of glycosidic nucleoside prodrugs. Among them, compound (**28**) (d4T-glucose-phosphodiesters) (Figure 8) was prepared by condensing glycosyl phosphoramidite 7 with the d4T nucleoside using an activating agent. This prodrug showed high anti-HIV-1 activity with an EC_50_ of 0.34 µM in PBM cells. This improved potency might be attributed to the presence of free nucleosides within the compound. Compound (**28**) showed negligible cytotoxicity in PBM cells (>100 µM) and Vero cells (>100 µM), and it had a modest IC_50_ value of 76.8 µM in CEM cells. This indicated that the conjugation performed did not fully stop the activity of the original d4T nucleoside, which exhibited an anti-HIV-1 activity with an EC_50_ value of 0.073 µM. Additionally, all prodrugs, including (**28**), exhibited higher solubility in water (average solubility of 16.7 mg/mL) than the original d4T compound (solubility of 5.4 mg/mL). Thus, prodrug (**28**) could enhance the oral bioavailability of nucleosides and improve their absorption in the intestine [78].

Mononucleotide prodrugs, also known as pronucleotides, derivatized from d4T, 2′,3′-dideoxythymidine (ddT), and 2′,3′-dideoxyadenosine (ddA), were synthesized and evaluated by Schlienger et al. (2022) [79]. To develop these prodrugs, two o-pivaloyl-2-oxyethyl substituents were integrated into a phosphorodithiolate structure, promoting biolabile phosphate protection [79]. Among these compounds, the derivative of ddA (compound (**29**)) (Figure 8) was found to be the most potent antiviral agent against HIV. Its EC_50_ value was 0.025 µM in CEM-SS cells, 0.27 µM in MT-4 cells, 0.02 µM in PBMCs, and 0.00023 µM in MDM cells [79]. This compound also had low cytotoxicity, with CC_50_ values greater than 10 µM in CEM-SS cells. In comparison, the original drug 2′,3′-dideoxyadenosine (ddA) exhibited less potent antiviral effects against HIV. Its EC_50_ values were 0.49 µM in CEM-SS cells, 7.8 µM in MT-4 cells, 0.24 µM in PBMCs, and 0.12 µM in MDM cells. Its CC_50_ values were greater than 100 µM in CEM-SS cells, MT-4 cells, PBMCs, and MDM cells [79]. Tests to evaluate the stability of the compound in nonenzymatic environments, such as phosphate buffer and RPMI, showed that the synthesized prodrug underwent minimal hydrolysis. The corresponding metabolites formed slowly, with half-lives of several days (t_1/2_ of 2.9 days in phosphate buffer at pH 7.4; t_1/2_ of 2.2 days in RPMI at pH 7.45) [79]. The prodrug was also converted more rapidly into its diester metabolite, indicating accelerated conversion that coincided with the intended release of mononucleotides in cells [79].

Gangadhara et al. (2014) synthesized a series of α- and β-carboxylated phospholipid prodrugs derived from dideoxy nucleosides (d4A, d4T, and ddC) and evaluated them against HIV-1-infected cells. The phospholipid scaffold increased lipophilicity and, thus, cellular penetration, while the α-carboxylated subunit contributed to the phosphodiester bond cleavage. The α-hydroxy D4T compound (**30**) (Figure 8) showed an IC_50_ value of 0.4 µM against both HIV-1 and HIV-2 in infected MT-4 cells, but it was inactive in CEM cells. In contrast, the β-hydroxy D4T compound (**31**) (Figure 8) was approximately 30-fold less active than (**30**), and both prodrugs were less active than stavudine (d4T). In the MT-4 cell model, the prodrug phosphodiester d4A-20, characterized by the presence of an α-hydroxy stearic acid d4A conjugate, showed an IC_50_ value of 0.19 µM and a CC_50_ value of 97.8 µM. In comparison, the reference compound, dideoxy nucleoside d4A, showed an IC_50_ value greater than 277.5 µM and a CC_50_ value of 277.5 µM. As these prodrugs showed high inhibitory ability, d4A-20 might undergo more facile enzymatic degradation in cells, which explains its superior effect [80].

A prodrug delivering nucleoside diphosphates and triphosphates directly inside cells can overcome inconveniences, such as fast metabolism or side effects, compared to the parental nucleoside [81]. Thus, methodologies to avoid such problems can help in evading enzymatic processes, such as nucleoside deamination or the dephosphorylation of nucleoside monophosphates. Researchers have developed some approaches to solve this issue by using cycloSal-technology [82] nucleoside diphosphate prodrugs (Di*PP*ro) and nucleoside triphosphate prodrugs (TriPPPro)—approaches [81,83,84]. 

CycloSal prodrugs have a cyclic salicylate moiety integrated into the prodrug structure [81,82,83,84]. It undergoes cleavage through chemically induced processes (Figure 9). This cyclic group serves as a carrier and can be enzymatically cleaved in vivo to release the active drug in its therapeutic form, which allows fine-tuning of release kinetics and targeting. The cyclic structure provides several benefits, such as high stability during circulation, protection against metabolic degradation, targeted delivery to specific tissues or cells, and the control of drug release that reduces the side effects [81]. 

Symmetric and nonsymmetric nucleoside diphosphate prodrugs (known as Di*PP*ro) can be used to perform high levels of selective delivery. Di*PP*ro exhibits HIV-1 antiviral activity, even in mutant thymidine-deficient (CEM/TK-) cells [85]. An advantage of asymmetric Di*PP*ro is that the levels of nucleoside monophosphate are also detected along with the levels of nucleoside diphosphate in the cellular extract. The level of nucleoside monophosphate was found to be correlated with the stability of the prodrug. This stability depended on the length of the alkyl residues denoted as “R”, which were attached via an ester linkage to the 4-hydroxybenzyl group [86]. Nucleoside monophosphate is formed due to a competitive reaction, where water or hydroxide ions interact with the phosphorus atom of the β-phosphate group, leading to the cleavage of the phosphate anhydride bond and the release of nucleoside monophosphate [86].

Weinschenk et al. (2015) designed nonsymmetric DiPPro-nucleoside diphosphate (NDP) prodrugs using active anti-HIV nucleoside analogs of d4T and AZT. Two different “masking units” were attached to the β-phosphate group of the respective nucleoside diphosphate [86]. The mechanism of delivery of these prodrugs was confirmed through hydrolysis studies. Experiments on the enzymatic hydrolysis of prodrugs, using cell extracts and pig liver esterase, led to the formation of nucleoside diphosphates. Additionally, PBS (phosphate-buffered saline) facilitated the production of nucleoside diphosphates and nucleoside monophosphates. Among the various compounds investigated, the DiPPro-d4TDPs compounds (**32–41**) (Figure 10) were prominent due to their unique composition. These compounds featured distinct aliphatic ((R_2_ = CnH2n+1) and aromatic (R_2_ = X-Ph) acyl moieties as lipophilic masks. Along with short aliphatic acyl moieties (R_1_ = CH_3_, C_4_H_9_) in the acyloxybenzyl-masking group, these nonsymmetric DiPP compounds exhibited high activity, often similar to or even higher than the activity of the original nucleosides in wild-type CEM cell cultures [79]. Specifically, the d4TDP derivatives (**32**–**41**), containing aliphatic ester functional groups in the acyloxybenzyl units, showed comparable activity against HIV-1 and HIV-2. This was observed within the same concentration range as that for the original d4T compound in wild-type CEM cell cultures. Compound (**39**) was found to be the most potent, with an EC_50_ value of 0.10 µM for HIV-1 and 0.28 µM for HIV-2. In contrast, D4T showed an EC_50_ value of 0.52 µM for HIV-1 and 2.23 µM for HIV-2 in CEM cell cultures. The observed antiviral efficacy was also recorded in the lipophilic compounds DiPPro (**33**) and (**38**) (Figure 10), as well as compound (**39**) (Figure 10), in thymidine (TK-)deficient mutant CEM cells infected with HIV-2. The antiviral activity was correlated with the lipophilicity of these compounds. This relationship occurred due to the introduction of acyl portions of the DiPPro prodrugs. These findings highlighted the efficient cellular uptake and intracellular delivery of a phosphorylated form of d4T (d4T diphosphate) [86].

Weinschenk et al. (2015) synthesized and evaluated DiPPro nucleotides, specifically a bis(benzyloxy benzyl)nucleoside diphosphate prodrug. Among these, the prodrug DiPPro-d4TDP, featuring a hydrogen donor at position 4 of the benzoyl segment (referred to as **42**) (Figure 10), showed higher activity than the original compound against both HIV-1 and HIV-2. In CEM/0 cells, its EC_50_ value was 0.40 µM for HIV-1 and 0.30 µM for HIV-2. The prodrug showed an EC_50_ value of 0.85 µM against HIV-2 in CEM/TK-cells. In comparison, unmodified compound d4T showed an EC_50_ of 0.86 µM for HIV-1 and 2.3 µM for HIV-2 in CEM/0 cells, as well as an EC_50_ of 173 µM for HIV-2 in CEM/TK-cells. Although the cytotoxicity of the prodrug was slightly high, with a CC_50_ of 36 µM in CEM/0 cells, the CC_50_ value of the original compound was greater than 250 µM. The chemical stability of the prodrug was assessed using PBS at pH 7.3 and CEM/0 cell extracts. It showed high stability, with a half-life (t_1/2_) of 82 h at pH 7.3 and a t_1/2_ of 7 h in CEM/0 cell extracts. An evaluation of the hydrolysis process and the role of enzymes indicated that the prodrug had a low retention time (tR) of 9.41 min in CEM/0 cell extracts, suggesting lower lipophilicity. Trace amounts of d4TMP were detected in this prodrug. These findings might be attributed to the dephosphorylation of d4TDP by phosphatases present in the cell extracts or the cleavage of the anhydride–phosphate bond after the hydrolysis of the benzoyl ester [87].

Schulz, Balzarini, and Meier (2014) synthesized and evaluated lipophilic d4T diphosphate prodrugs using the Di*PPro* approach. Convergent synthesis was performed with the protection of the β-phosphate group of nucleoside diphosphates as bis(acyloxybenzyl)phosphate esters. The (**43**) ((*N*[nBu]4)-Bis-(4-decanoyloxybenzyl)-d4TDP) compound (Figure 10) exhibited an EC_50_ value of 0.080 µM for HIV-1 and 0.32 µM for HIV-2 in CEM cells, 0.11 µM for HIV-2 in CEM/TK^−^, and a CC_50_ value of 62 µM. The parent drug d4T showed an EC_50_ value of 0.86 µM for HIV-1 and 2.3 µM for HIV-2 in CEM cells, 173 µM for HIV-2 in CEM/TK^−^, and a CC_50_ value >250 µM [81]. Compared to the parent nucleoside d4T, this antiviral activity was 1570 times greater in TK-deficient CEM cells; the prodrug was more active in infected wild-type CEM cells. To assess the effects of the counterion, C9-DiPProd4TDP **4** was saltified with ammonium and tetra-n-butylammonium [81]. Cytotoxicity and antiviral activity remained unaffected. In studies on stability conducted in aqueous 25 mM phosphate buffer (PBS; pH 7.3), the T_1/2_ for the initial hydrolysis process was 63 h with the formation of d4TDP, and the t_1/2_ of the second step was 280 h, with the formation of d4TMP. These findings showed that by making the prodrugs more lipophilic than the compounds synthesized earlier, the Di*PPro* method was improved [81].

Zhao et al. (2020) performed molecular modification of the analogs of d4TTP (the triphosphate form of d4T) involving lipophilic γ-alkyl groups and acyloxybenzyl prodrugs, to deliver γ-alkyl-d4TTP into cells. After synthesis, the effectiveness of those prodrugs was assessed against HIV-1 and HIV-2 infected cells. The γ-alkyl-nucleoside triphosphates were highly stable under enzymatic dephosphorylation treatment in cell extracts. They remained stable for a minimum of 30 h. These modified compounds exhibited high antiviral efficacy against both HIV-1 and HIV-2 in CD4+ T-lymphocyte CEM cell cultures (CEM/0). Additionally, they were effective against HIV-2 in cell cultures deficient in thymidine kinase (CEM/TK-cells), which indicated that they could pass through the cell membrane. Among these compounds, γ-C18-alkyl-d4TTP (**44**) (Figure 10) was the most potent in CEM/0, showing an EC_50_ value of 0.11 µM against HIV-1 and HIV-2. Its activity against HIV-2 in CEM/TK-cells was especially high, with an EC_50_ value of 0.05 µM, representing a 2700-fold increase in potency compared to d4T. These compounds were found to exclusively function as substrates for HIV-RT in primer extension assays, and they were not recognized by DNA-polymerases α, β, or γ. Thus, these prodrugs with high selectivity toward viral polymerase are promising agents for efficaciously delivering nucleoside triphosphates [88].

Due to the negative charge, nucleoside triphosphate (NTP) demands optimization using a prodrug approach. NTPs containing reversible modifications due to the inclusion of the γ-phosphate were synthesized and evaluated by Gollnest et al. (2015). These modifications included two lipophilic masking units at the γ-phosphate and d4T. These modifications were aimed at augmenting the limited phosphorylation found in nucleoside reverse transcriptase inhibitors [89]. Compound (**45**) (Figure 10), c-Bis-(4-octadecanoyloxybenzyl)-d4TTP (R = C_17_H_35_), characterized by two identical 4-alkanoyloxybenzyl groups, showed higher antiviral efficacy against HIV than the original d4T compound. This increase was attributed to its lipophilicity, facilitating cellular permeation. In mesenchymal stem cell (MSC) cells, the EC_50_ value was 0.17 µM for HIV-1 and 0.31 µM for HIV-2. In MSC/TK-cells, it showed an EC_50_ value of 0.28 µM for HIV-2 and a CC_50_ value of 29 µM in MSC cells. In contrast, the original d4T compound exhibited an EC_50_ value of 0.33 µM against HIV-1 in CEM cells, 0.89 µM against HIV-2 in CEM cells, and 150 µM against HIV-2 in CEM/TK-cells. It also showed a CC_50_ value of 79 µM in CEM cells. The evaluation of the hydrolysis of the prodrug in PBS at pH 7.3 showed that its half-life for the first removal of one masking unit (t_1/2_ (1)) to generate the intermediate was 50 h. For the second hydrolysis step (t_1/2_ (2)) leading to the formation of triphosphate (d4TTP), the half-life observed was 583 h. Enzymatic hydrolysis using cell extracts (CEM cells) showed a t_1/2_ (1) of 13 h. These findings highlighted the occurrence of intracellular enzymatic hydrolysis, directly leading to the intracellular formation of phosphorylated d4T metabolites, specifically d4TTP in this case [89].

The TriPPPro approach is an extremely promising strategy in the field of nucleotide prodrugs, as this method has a unique ability to penetrate cell membranes without relying on cellular kinases for activation. In the nucleoside triphosphate delivery system, the terminal γ-phosphate group is concealed using a pair of lipophilic acyloxybenzyl (AB) groups that are susceptible to enzymatic cleavage [89]. The high permeability of such prodrugs leads to their accumulation in thymidine kinase-deficient CEM/TK(-) cells [84]. 

Prodrugs of d4T nucleoside triphosphate (NTP) analogs were obtained by investigating the TriPPro approach. For this, Jia, Schols, and Meier (2020) incorporated distinct biodegradable masking subunits along with an acyloxybenzyl (AB) moiety into the γ-phosphate group of the compound [83]. Using this method, compounds (γ-ACB;AB-d4TTP) and (γ-(ACB;ACB)-d4TTP) were synthesized. Among these compounds, prodrug (**46**) (R_1_: C_2_H_5_; R_2_: C_16_H_33_) (Figure 10) exhibited high antiviral activity against HIV-1 and HIV-2 in wild-type CEM/0 cell cultures [90]. It was effective, considering that it had an EC_50_ of 0.027 µM for HIV-1 and an EC_50_ of 0.0048 µM for HIV-2. In comparison, the reference compound d4T showed EC_50_ values of 0.43 µM against HIV-1 and 0.31 µM against HIV-2. Prodrug (**46**) showed higher activity against HIV-2 (EC_50_: 0.11 µM) in mutant CEM/TK-cells due to its high lipophilicity. In contrast, the reference d4T showed a substantially higher EC_50_ value (31.05 µM) against HIV-2 under the same conditions [90]. The chemical stability of these compounds was assessed in PBS at pH 7.3. Prodrug **46** (C2-AB; C16-ACB) showed high stability with a half-life (t_1/2_) of 83 h; in contrast, the reference d4TTP had a t_1/2_ greater than 500 h. When the TriPPPro composite (**46**) was hydrolyzed, the researchers found a specific cleavage of the biodegradable moiety, which resulted in the formation of an intermediate known as γ-(C16-ACB)-d4TTP or (**46i**) (Figure 8). This intermediate (**46i**) initially accumulated and underwent subsequent cleavage to finally yield d4TTP. In hydrolysis assays involving exposure to human CD4+ T-lymphocyte cell extracts, prodrug (**46**) (with a t_1/2_ of 1.9 h) displayed efficient cleavage, which led to the production of the intermediate γ-(C16-ACB)-d4TTP (**46i**). These findings collectively elucidated the asymmetric TriPPPro concept, where the γ-phosphate of the NTP was modified in a bioreversible manner. This mechanism facilitated the targeted delivery of d4TTP with high selectivity through an enzyme-activated pathway. Using this strategy, the intracellular phosphorylation steps can be avoided [90].

While synthesizing γ-nonsymmetrical nucleoside analog triphosphates through the TriPPPro method, Zhao et al. (2021) incorporated two distinct AB masks, which were attached to the γ-phosphate group [91]. The evaluation was conducted in CD4+ T-lymphocyte CEM cell cultures. The compounds, denoted as γ-(AB, ab)-d4T, showed high efficacy against HIV-2 and comparable or slightly higher performance against HIV-1 compared to the parent nucleosides d4T or d4TTP [91]. Among these compounds, the effect of compound (**3ce**) was prominent. This compound is a γ-(AB-iso-C_4_H_9_,ab-C_14_H_29_)-d4TTP ammonium salt. It exhibited high potency, with an EC_50_ value of 0.17 µM against HIV-1 in CEM cells and the same EC_50_ value against HIV-2. The CC_50_ value was 24 µM. Compound (**47**) (Figure 10) exhibited a 128-fold increase in activity compared to the parent compound d4T. However, the newly synthesized compounds discussed in this study had slightly lower potency than the Tri*PP*Pro compounds containing two alkylacyl AB-masks [91]. Assessment of the chemical stability of compound (**47**) (characterized by a branched iso-butyl group) in an aqueous solution of 25 mM phosphate buffer (PB) at pH 7.3 showed a half-life (t_1/2_) of 64 h. Additionally, the principal hydrolysis products identified included the original drugs (d4TTP and d4TDP) and small quantities of d4TMP (with concentrations less than 4%) [91].

### 2.5. Emtricitabine Prodrugs

Emtricitabine (FTC) is a nucleoside transcriptase reverse inhibitor administered orally once a day; the (-) enantiomer is more active than its antipode. It is used in combination with antiviral drugs for HIV treatment, as well as in preexposure prophylaxis (PrEP). The drug has a relatively long half-life, and its main adverse effects include headache, nausea, diarrhea, and fatigue. In some cases, it can cause changes in kidney function and bone density. Lactic acidosis and hepatomegaly with steatosis are rare events [92]. 

Extended-release injectable (ERI) formulations have been investigated for the treatment of several diseases, including AIDS, as they can address challenges linked to medication nonadherence. For HIV treatment, the approval of ERI containing rilpivirine and cabotegravir provided a formulation that allowed therapeutic plasma concentrations of the drug for up to 20 weeks [93,94]

Three FTC prodrugs (**48–50**) (Figure 11) were synthesized by Curley et al. (2023), where they were incorporated into injectable aqueous semisolid prodrug nanoparticles (SSPNs) [94]. The pharmacokinetics of SSPNs containing prodrugs were evaluated over 28 days through intramuscular injection in Wistar rats, New Zealand white rabbits, and Balb/C mice. Two formulations were selected to evaluate their potential to prevent HIV infection in NSG-cmah−/− humanized mice, based on the study of preformulations. After injection, the peak concentration (Cmax) was reached by 12 h in rats, 48 h in rabbits, and 24 h in mice. Plasma concentrations fell below the detectable threshold of 2 ng/mL by day 21 in rats and rabbits and by day 28 in mice. Even 28 days after infection, HIV RNA remained undetectable in the plasma, spleen, lung, and liver samples [94]. 

Targeting the primary reservoir of HIV lymphoid tissues and infected cells is an enormous challenge that needs to be solved to eliminate the virus and achieve a functional cure in the next few years [95]. Class A scavenger receptors (SR-A), which are expressed in myeloid type cells (i.e., monocytes, macrophages, and dendritic cells), might be used to selectively deliver the drug to those cells, which can decrease undesirable systemic effects [89]. Endothelial scavenger receptor-A (SR-A) facilitates the transcytosis of modified LDL in the lymphatic system. This provides an opportunity for targeted delivery via SR-A to the lymphatic system [96].

The FTC prodrug, known as PLS-FTC (**51**) (Figure 11), was designed by Stevens et al. (2020) through esterification by conjugating it with succinylated poly(L-lysine) (PLS) to increase its distribution in the lymphatic system. Experimental results for compound (**51**) showed a release half-life of 15 h in human plasma and 29 h in plasma with inhibited esterase. After incubation for 24 h in peripheral blood mononuclear cells (PBMCs), the drug released from (**51**) was converted into the active metabolite FTC triphosphate. The ratio of active metabolite to precursor (FTC-TP/FTC) was 1.3 compared to the initial ratio of 0.4, which indicated that the conversion of the prodrug was more efficient. In pharmacokinetic assessments conducted in rats, (**51**) concentrations were 7–19 times higher than FTC concentrations in lymphatic tissues [90]. This finding emphasized the enhanced distribution potential of (**51**) in lymphatic tissues, highlighting that this macromolecular platform might be effective [97].

Agarwal et al. (2013) prepared and evaluated FTC-based ester conjugates against cell-free and cell-associated HIV-1 viruses [98]. This myristoylated FTC conjugate, referred to as compound (**52**) (Figure 11), exhibited high lipophilicity with a log *p*-value of 5.96, whereas for FTC, the calculated value was −1.29, which limited its permeability and its effective internalization by cells. At the highest tested concentrations, prodrug (**52**) showed negligible cytotoxicity (CC_50_ > 200 µM). This myristoylated conjugate showed high anti-HIV activity. Against cell-free viruses, it exhibited an EC_50_ of 0.07–0.1 µM, which was approximately 10–19 times greater than the EC_50_ of FTC alone (EC_50_ = 0.7–1.9 µM). Compound (**52**) showed an EC_50_ of 3.7 µM against cell-associated viruses and outperformed FTC by approximately 24-fold (EC_50_ = 88.6 µM). When evaluated against clinical HIV isolates, prodrug (**52**) demonstrated potent anti-HIV activity against clade B and C clinical isolates, with 90% inhibitory concentration (IC_90_) values of 6.6 and 10.9 nM, respectively. These values were considerably higher than those of FTC (IC_90_ = 32.4 and 161.9 nM), exhibiting a 5-fold to 15-fold increase in potency. Additionally, when tested against drug-resistant viral mutations, specifically those resistant to nonnucleoside reverse transcriptase inhibitors (B-NNRTI) and those exhibiting TDF-NRTI resistance (B-K65R), prodrug (**52**) exhibited IC_90_ values of 15.7 nM and 16.1 nM, respectively. This presented a substantial decrease in potency compared to FTC (IC_90_ = 103 and 567 nM), with reductions of 6.6-fold and 35.2-fold, respectively. Cellular uptake studies in which human T-lymphoblastoid cells were exposed to prodrug (**52)** at 50 µM for 1–24 h at 37 °C confirmed through HPLC analysis that prodrug (**52**) was internalized within 1 h of incubation (retention time: 18.2 min). However, at the 12th hour, the intracellular levels of prodrug (**52**) decreased, suggesting hydrolysis to FTC catalyzed by intracellular esterases. The HPLC profiles also indicate that intracellular hydrolysis occurred, and potentially phosphorylated products were present [98]. Prodrug (**52**) disappeared from cell extracts after 24 h, with a distinct peak indicating the presence of metabolic products at 1.7–2.9 min [98].

NTP prodrugs can considerably enhance cellular uptake by infected cells. The antiviral activity profiles against HIV-1 and HIV-2 displayed substantial variations in potency, depending on the lipophilicity of the NTP prodrugs. Studies using prodrugs of emtricitabine (FTC), denoted as (**53**), were conducted on cultures of infected wild-type CD4+ CEM T-cells and thymidine kinase-deficient CD4+ T-cells (CEM/TK−). The research carried out by Jia, Schols, and Meier (2020), with NTP prodrug (**53**) (Figure 11) (with R1: C_4_H_9_ and y-(C4-AB; C16-ACB)), showed high antiviral effectiveness against HIV-2 CEM/TK−. This effectiveness was indicated by a lower EC_50_ of (**53**) (EC_50_ = 0.029 µM) compared to that of its parent nucleoside FTC (EC_50_ = 0.046 µM). For infected wild-type CD4+ CEM T-cells, the prodrug (**53**) exhibited a robust HIV-1 EC_50_ of 0.0043 µM and an HIV-2 EC_50_ of 0.0087 µM. The prodrug outperformed the reference compound FTC with EC_50_ values of 0.010 µM for HIV-1 and 0.016 µM for HIV-2. When the stability of the prodrug (**53**) in cell extracts was assessed via hydrolysis, it showed a considerably rapid formation of nucleoside analog triphosphates in human T-lymphocyte CD4+ CEM cell extracts. Their half-lives ranged from 1.1 h to 5 h, independent of the specific attached nucleoside [90]. This finding suggested that an enzymatic cleavage occurred. Compound (**53**) underwent rapid hydrolysis, generating the corresponding NTPs in concentrations similar to those recorded in other stability assays (CEM/0 t_1/2_ 2.78 h). This confirmed the release of NTPs in biological media, such as T-lymphocyte extracts. These findings indicated that FTC prodrugs can be further modified to improve the effects of nucleoside analogs into potent biologically active metabolites [90].

### 2.6. Abacavir (ABC) Prodrugs

The presence of HIV in critical locations, such as the central nervous system, poses a significant challenge to antiretroviral therapy, as penetrating the blood–brain barrier is difficult. This issue contributes to the maintenance of the HIV-1 reservoir. P-glycoprotein (P-gp), a member of the ATP-binding cassette (ABC) family of multidrug resistance transporters, is a major contributor to this limitation. P-gp is highly expressed in brain capillaries, which further hinders access to antiretroviral drugs. Strategies involving P-gp inhibition might be effective in eradicating the HIV reservoir. One such strategy involves ABC, a P-gp substrate. The ABC dimers synthesized by Namanja et al. (2012) were evaluated by inhibiting P-gp efflux at the blood–brain barrier and reverting to monomeric therapeutic forms under reducing conditions. ABC dimers were evaluated in vitro using a P-gp overexpressing the T lymphoblastoid cell line (12D7-MDR) via a flow cytometry assay to determine their inhibitory effect on P-gp-mediated transport of the fluorescent substrates calcein-AM and 4-fluoro-7-nitro-2,1,3-benzoxadiazole-abacavir (NBD-ABC). All dimers strongly inhibited P-gp efflux for the fluorescent substrates. Among these, (**54**) (Figure 12), characterized by the linkage of two monomers via an ester containing a disulfide and featuring four additional methyl units adjacent to the ester carbonyl in the tethers (represented as Me4), showed the highest activity. Its IC_50_ values were 0.6 µM for calcein-AM, 0.7 µM for NBD-ABC, and 65 nM for [^125^I] IAAP competition. Compound (**54**) exhibited an extended half-life of over 100 h in human plasma and 17.2 h in dithiothreitol (DDT) [99]. This significant improvement in stability, which was approximately fourfold greater than ABC, indicated the reversion rate of abacavir prodrug dimers. Compound (**54**) was inactive in an in vitro reverse transcriptase (RT) assay, suggesting that its cellular antiviral activity was linked to the reversion of the prodrug dimer to the RT-active monomeric parental drug ABC. Thus, the researchers inferred that the conversion of ABC from the dimer form to the therapeutic form in the reducing environment of the cell was responsible for its antiviral activity. This mechanism can be used to modify the release rate of the abacavir monomer from the prodrug [99].

Weising et al. (2018) synthesized and evaluated lipophilic NTP-ABC (triphosphate prodrugs of abacavir), along with their 1′,2′-cis-disubstituted analogs, in T-lymphocyte cell cultures (CEM cells) infected with either HIV-1 or HIV-2 [93]. Compared to their respective parent compounds, prodrug (**55**) (NTP-ABCTP, ammonium salt) (Figure 12) showed 4.5 times higher activity against HIV-1. Prodrug (**55**) showed EC_50_ values of 1.3 µM for HIV-1 and 1.4 µM for HIV-2 in CEM/0 cells. It also showed an EC_50_ of 2.1 µM for HIV-2 in CEM TK-cells and a CC_50_ of 60 µM in CEM T-cell cultures. In comparison, the parent drug monophosphate ABC showed EC_50_ values of 5.9 µM for HIV-1 and 5.2 µM for HIV-2 in CEM cells. It also exhibited an EC_50_ of 7.0 µM for HIV-2 in CEM TK-cells and a CC_50_ value of 135 µM in CEM T-cell cultures. No activity of 1′,2’-cis substituted nucleoside was detected. The provided triphosphates were not eliminated, and the compounds could not be rephosphorylated from their monophosphate or diphosphate forms. Even if triphosphates are produced, HIV reverse transcriptase may not use them as substrates. This aspect might be further investigated by conducting primer-extension tests using reverse transcriptase and separated triphosphates [100].

The compound (**56**) (Figure 12), a linked heterodimeric compound consisting of NRTI ABC and the protease inhibitor nelfinavir, was synthesized by Agrawal et al. (2020) as a prodrug to address the therapeutic evasion of P-glycoprotein (P-gp) through a prodrug strategy involving a combination of dual modes of action. This mutual prodrug could strongly inhibit HIV P-gp efflux, even in human brain endothelial cells. The linkage between ABC and nelfinavir was established via an ester linkage with a disulfide-containing ether. In the cellular reducing environment, the central disulfide is reduced, generating thiols that rearrange to release the monomeric drugs. The compound (**56**) showed high anti-HIV activity, as determined by the p24 protein levels in 12D7 cell lines infected with HIV-1LAI. At concentrations of 0.08, 0.31, and 1.25 µM, (**56**) decreased the content of p24 (<100 pg/mL) to levels less than that recorded in the control (>500 pg/mL and <1000 pg/mL) and similar to that recorded after treatment with a 1:1 mixture of ABC and nelfinavir (<100 pg/mL of p24). Compound (**56**) also showed a potent dose-dependent inhibition of P-gp efflux in assays involving the fluorescent substrates calcein-AM and NBD-ABC. This was found in 12D7-MDR cells (CD4+ T lymphocytes with P-gp overexpression) and hCMEC/D3 cells (immortalized human brain capillary endothelial cells expressing P-gp). The IC_50_ values for (**56**) were submicromolar, ranging from 0.41 to 0.77 µM for different substrates and cell types. In comparison, nelfinavir had higher IC_50_ values (ranging from 1.7 to 9.1 µM), indicating that its inhibitory effect was less potent. The results of 3-[4,5-dimethylthiazol-2-yl]-2,5 diphenyl tetrazolium bromide (MTT) assays showed that (**56**) maintained cell viability (>95%) without inducing toxicity at its maximum tested concentration (20 µM). In experiments that simulated a reducing environment, where disulfide bonds within the heterodimers yielded thiols that rearranged into monomeric drugs (ABC and nelfinavir), the regeneration process was fast (full reduction in 1 h). The half-life (t_1/2_) of ABC regeneration from the heterodimer (**56**) was 31.8 h with dithiothreitol (DTT) and 29.4 h with glutathione (GSH). Nelfinavir regeneration had a t_1/2_ of 1.2 h with DTT and 10.3 h with GSH, releasing faster than ABC, due to the phenolic moiety’s better leaving group ability in nelfinavir within (**56**) [101].

### 2.7. Others

Membrane-associated drugs can profoundly influence the pharmacokinetic, safety, and efficacy attributes of medicinal agents. This approach is particularly valuable in drug discovery, especially for compounds facing pharmacokinetic challenges. Many membrane transporters can perturb the internal movement of bioactive substances and alter their absorption and biodistribution; these transporters include members of the ATP-binding cassette (ABC) and solute carrier (SLC) transporter superfamilies. For example, certain SLC transporters, such as OAT1 (SCL22A6), act as selective substrates for compounds such as adefovir, ciclofovir, zidovudine, lamivudine, and tenofovir. These substrates are present in organs such as the placenta and the proximal tubule of the kidneys. They affect drug disposition and elimination and thus play a role in drug–drug interactions [102,103].

One of the best examples of drug designs exploring the use of membrane transporters to increase bioavailability is the antiviral valacyclovir. This drug is commonly prescribed for treating genital herpes (herpes simplex virus type 2), cold sores (herpes simplex virus type 1) in adults, and shingles (herpes zoster). It can reduce the severity of herpes outbreaks in individuals with a weak immune system, such as those living with HIV/AIDS [104]. Valacyclovir exhibits 3–5 greater oral bioavailability (~55%) than its parental drug acyclovir, which is attributed to its recognition and transport by PepT1 [105,106,107].

Enhancing the ability of antiretroviral drugs to penetrate the CNS can decrease the severity of HIV-1 infection in the brain and mitigate the cognitive impairments linked to it [108,109]. One example of this mechanism is the transporter called breast cancer resistance protein (BCRP), which is expressed in human brain microvessel endothelial cells and mouse brain capillaries. For this transporter, Abcg2^−/−^ knockout mice exhibited elevated levels of abacavir in the brain, suggesting the in vivo participation of BCRP in transporting antiretroviral drugs across the blood–brain barrier (BBB) [110]. Another example includes the transporter MRP1 involved in the entry of emtricitabine inside lymphocytes. Mrp2^−/−^ deficient rats showed a considerable decrease in the hepatobiliary elimination of TDF [111,112]. 

Investigating membrane transporters for designing drugs is a difficult task for HIV therapy, as many factors can influence the expression of transporters responsible for the influx and efflux of drugs [113] Efflux transporters, such as P-gp and BCRP, which are highly expressed in the apical surface of enterocytes can be influenced by protease inhibitors (e.g., ritonavir and darunavir). An increase in the level of TDF in the bloodstream was also associated with the inhibition of P-glycoprotein (P-gp) due to concurrently administered protease inhibitors, such as atazanavir, lopinavir, ritonavir, and darunavir [114]. 

Intestinal permeability can also be altered by food intake, inflammatory status, and the presence of viral proteins (e.g., Tat, gp120). High levels of proinflammatory cytokines and oxidative stress can also regulate ABC transporters [115,116] These findings showed that ABC drug transporters might be modulated by inflammation and oxidative stress linked to HIV-1, potentially leading to changes in the distribution of antiretroviral drugs in different organs [103]. 

Utilizing drug delivery systems can improve drug transportation while preventing the concurrent inhibition of multiple transporters, a situation that might otherwise lead to harm by enabling the permeation of toxins. Pharmacologically modifying specific transporters by simultaneously administering ART and transporter-targeted inhibitors is an effective strategy for elevating levels of anti-HIV drugs in tissues [108]. Miller et al. (1997) reported that micelles of pluronic copolymers can inhibit P-gp and can thus influence drug transport [117]. Incorporating antiviral drugs (e.g., AZT and 3TC) with pluronic copolymers into human monocyte-derived macrophages infected with HIV-1 resulted in a considerable reduction in the percentage of HIV-1-infected monocyte-derived macrophages (8–22% of the control). The antiviral impact of this modification exceeded that of the antiretroviral drugs, reaching 38% of the control [118].

Function, distribution, and transport potency need to be further studied to design new therapeutic antiviral drugs. New prodrugs need to be designed to enhance the passage of drugs through blood–testicular, blood–mucosa, blood–cerebrospinal fluid, and blood–brain barriers. 

## 3. Conclusions and Outlook

Although HIV treatment has progressed considerably, several limitations and challenges still need to be addressed, including therapy nonadherence, long-term toxicity, improvements in drug efficacy, and elimination of the viral reservoir. The prodrug approach is a promising tool to address these main drawbacks, particularly for overcoming the problems related to NRTIs. An example of the successful utilization of the prodrug approach is TDF alafenamide, which received FDA approval in 2016. 

In this review, we focused on NRTIs and presented several illustrative examples demonstrating the promising application of prodrugs to enhance solubility, bioavailability, pharmacokinetics, and drug delivery. Challenges related to targeting specific tissues/organs, such as the central nervous and lymphatic systems, have been effectively addressed through this approach. We highlighted several perspectives for designing prodrugs to treat neuroAIDS, where inadequate NRTI levels in the CNS are a common challenge. We also discussed approaches to reduce HIV reservoirs by investigating ways to deliver drugs to the lymphatic system. The combination of the prodrug approach with suitable formulations is an effective strategy to enhance the efficacy and safety of treatment. Understanding the function and distribution of membrane transporters might help deliver and accumulate drugs in desired compartments. In this review article, we only highlighted promising strategies that should be pursued to advance therapy in the future.

## Figures and Tables

**Figure 1 viruses-15-02234-f001:**
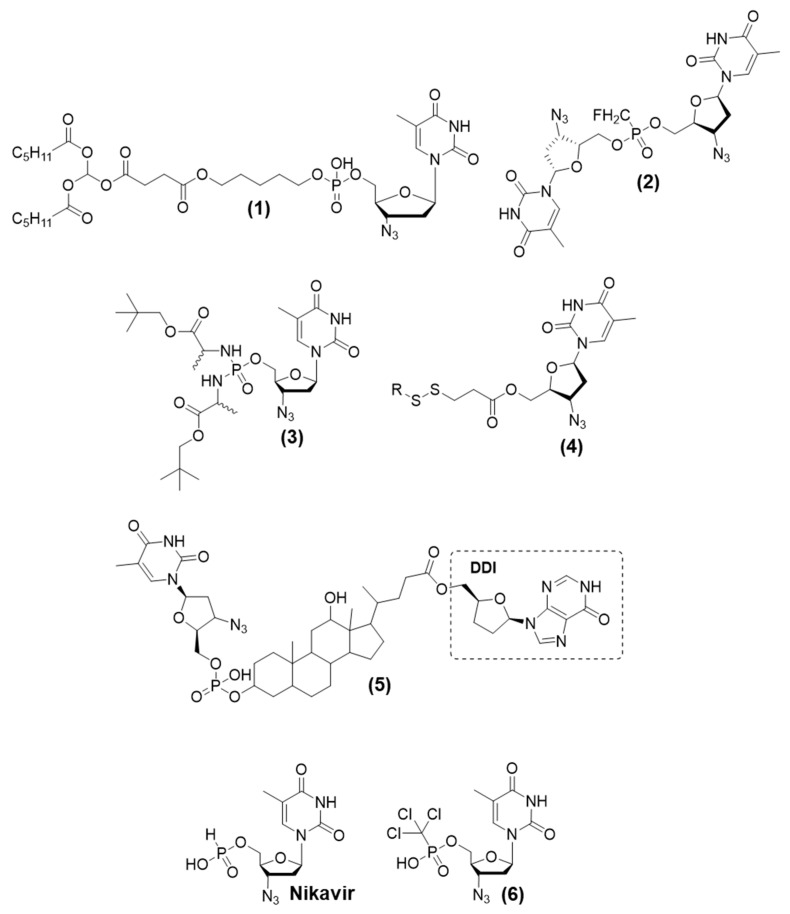
Structural representation of zidovudine prodrugs.

**Figure 2 viruses-15-02234-f002:**
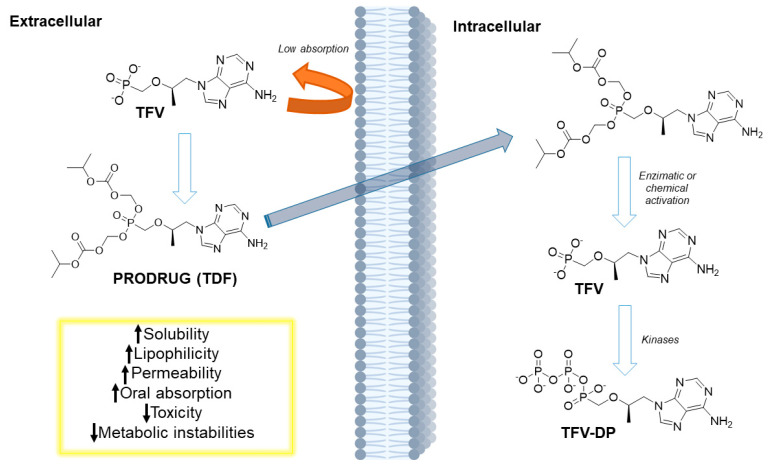
A diagrammatic representation of the prodrug approach to improve TFV properties. TDF can improve pharmacokinetic properties, as it can cross the biological membrane. Enzymatic or chemical activation leads to a TFV form that is then converted to TFV-DP by the action of the kinases.

**Figure 3 viruses-15-02234-f003:**
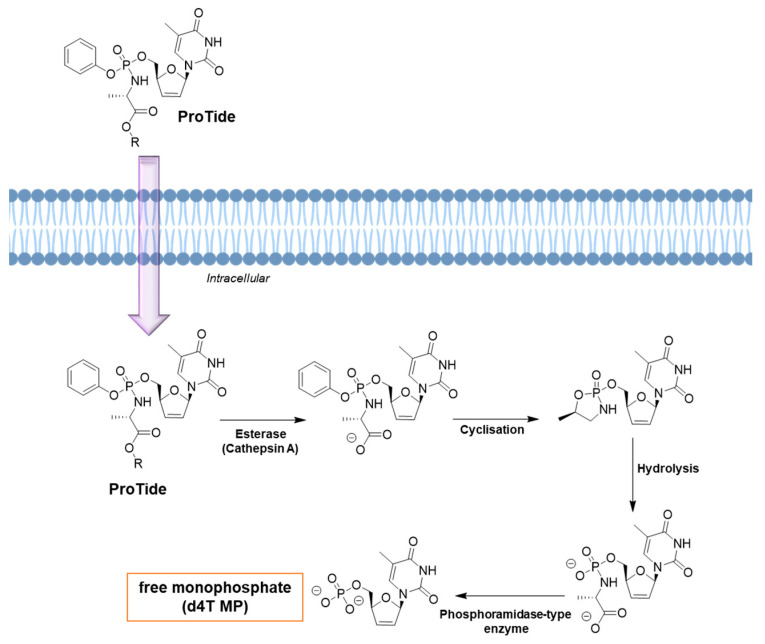
The proposed mode of action of ProTides. ProTide can efficiently cross the cell membrane and is enzymatically cleaved off in the intracellular region, releasing the free nucleoside monophosphate. TVF and TAF are examples of the ProTide approach.

**Figure 4 viruses-15-02234-f004:**
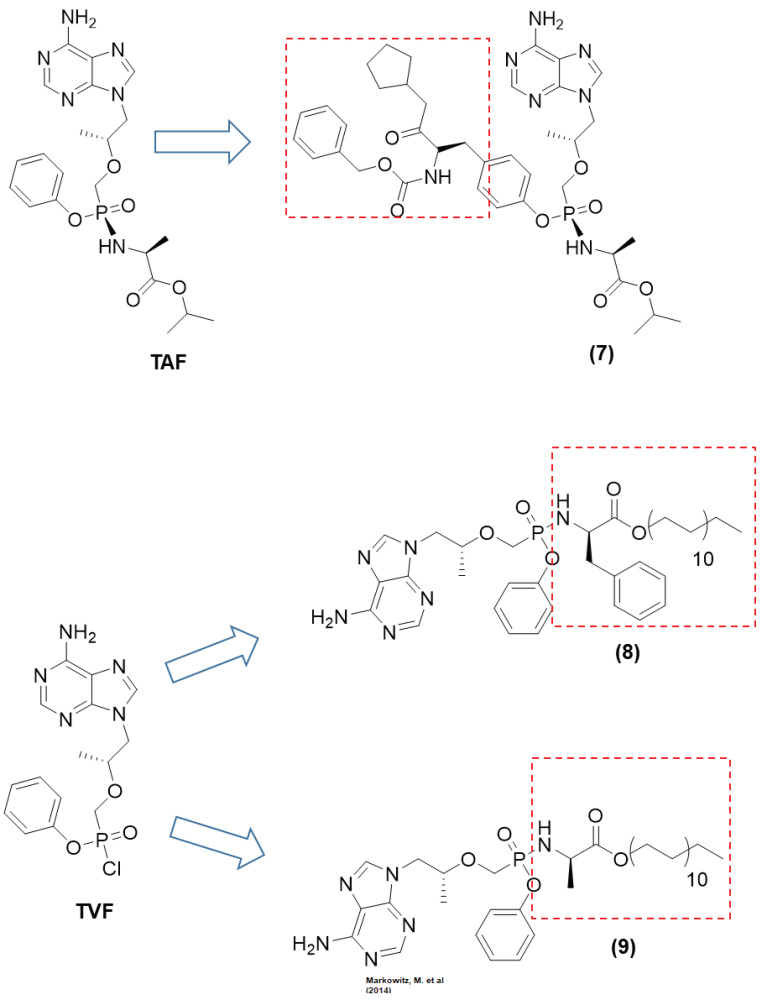
Structural modifications led to the obtaining of derivatives of TAF (**7**) and TVF (**8** and **9**). Red dots represents lypophilic subunits [39].

**Figure 5 viruses-15-02234-f005:**
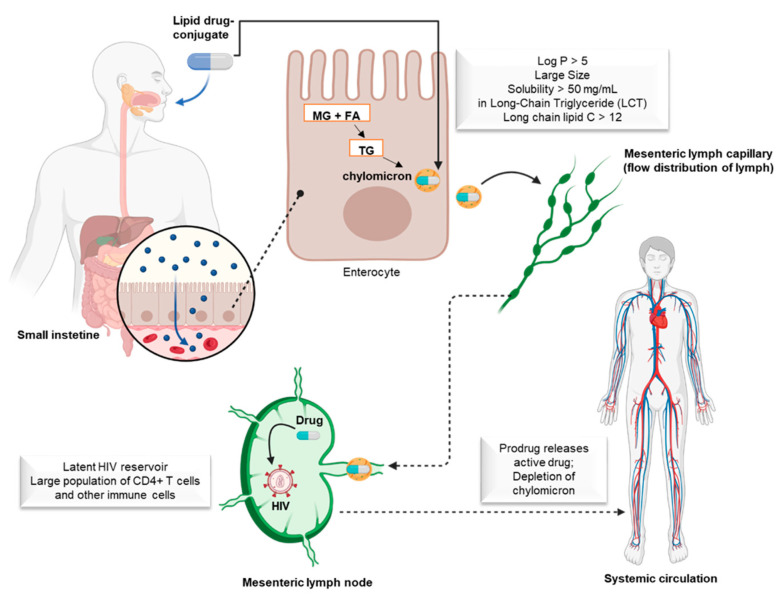
Lipid–drug conjugate metabolism. Lipophilic drugs exert their effects until they reach systemic circulation.

**Figure 6 viruses-15-02234-f006:**
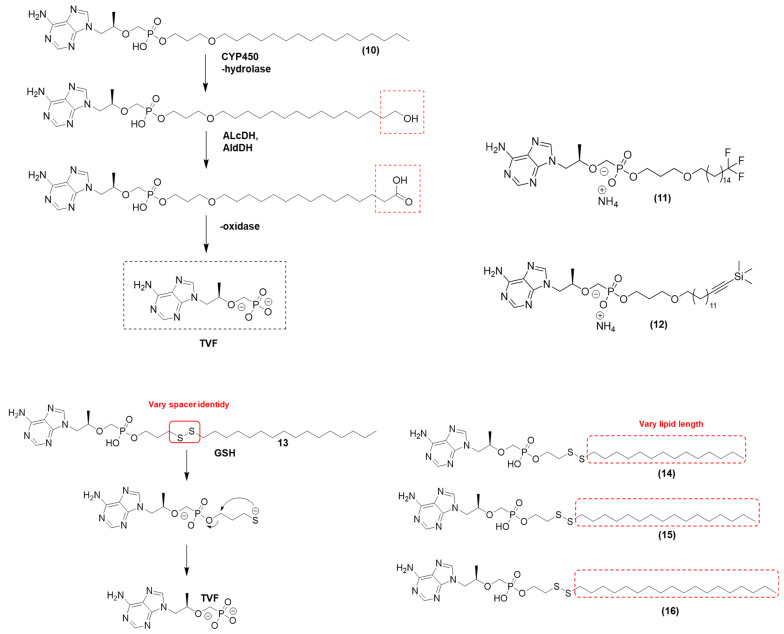
Structural representation of tenofovir prodrugs. Red dots highlights some of the modifications performed in the drugs.

**Figure 7 viruses-15-02234-f007:**
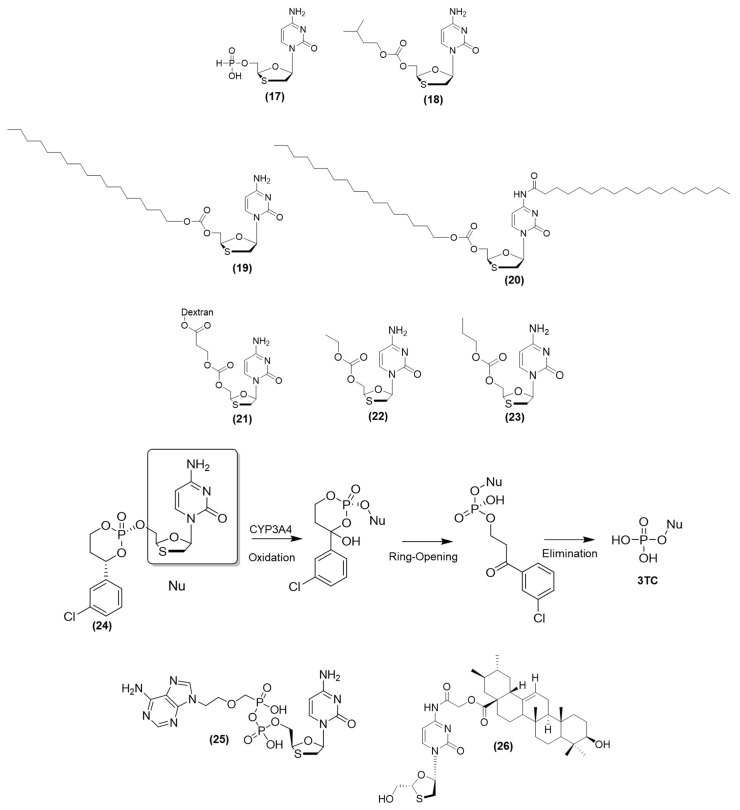
Structural representation of lamivudine prodrugs.

**Figure 8 viruses-15-02234-f008:**
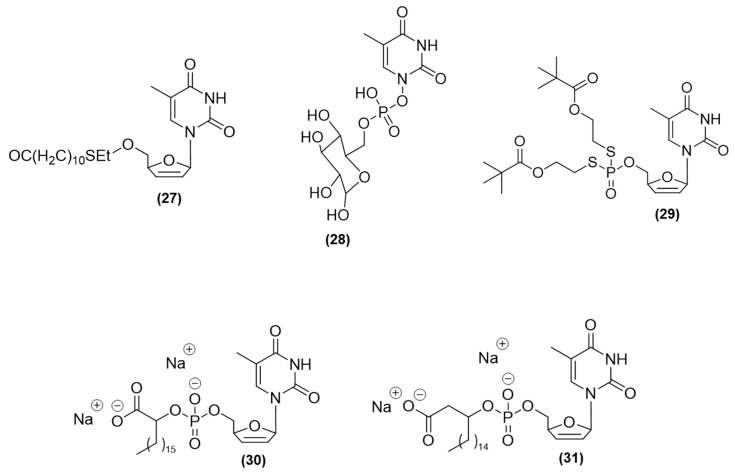
Structural representation of stavudine prodrugs.

**Figure 9 viruses-15-02234-f009:**
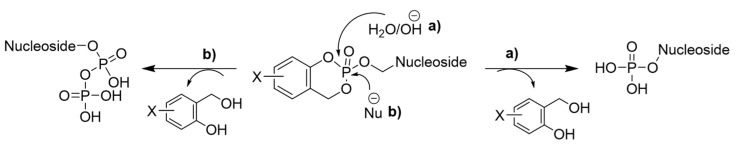
Drug release mechanism of cycloSal prodrugs. The cyclic group undergoes cleavage through chemically induced processes in two different regions as presented by (a) and (b).

**Figure 10 viruses-15-02234-f010:**
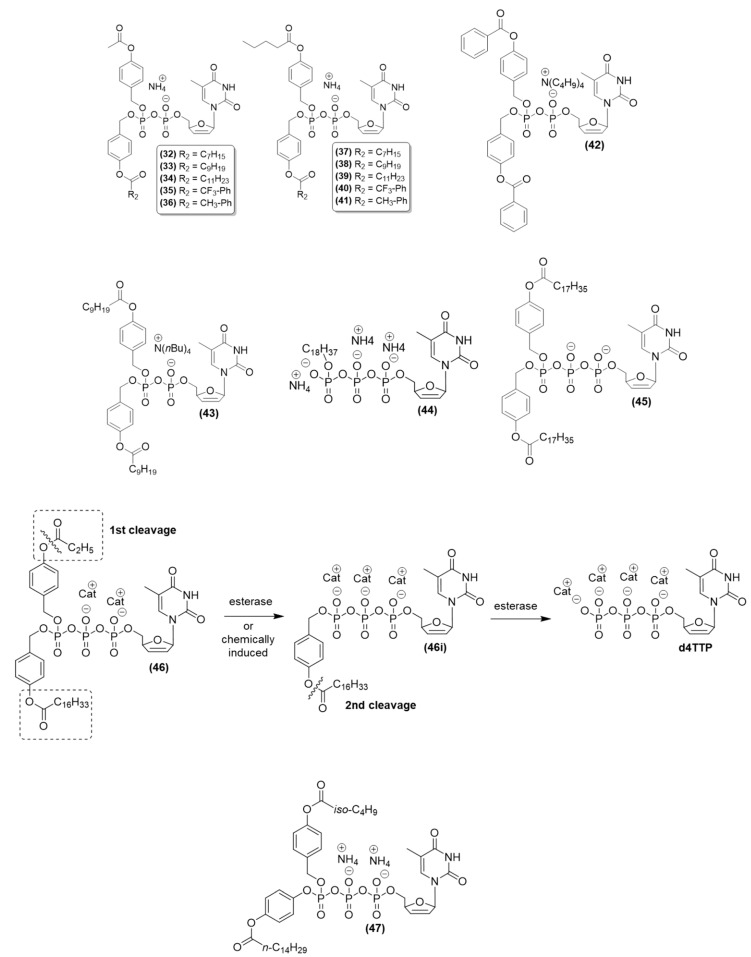
Structural representation of stavudine prodrugs, using the Di*PP*ro and TriPPPro approaches.

**Figure 11 viruses-15-02234-f011:**
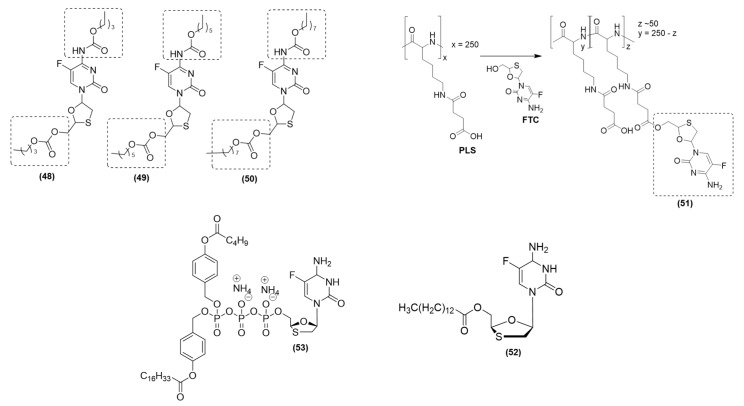
Structural representation of emtricitabine prodrugs.

**Figure 12 viruses-15-02234-f012:**
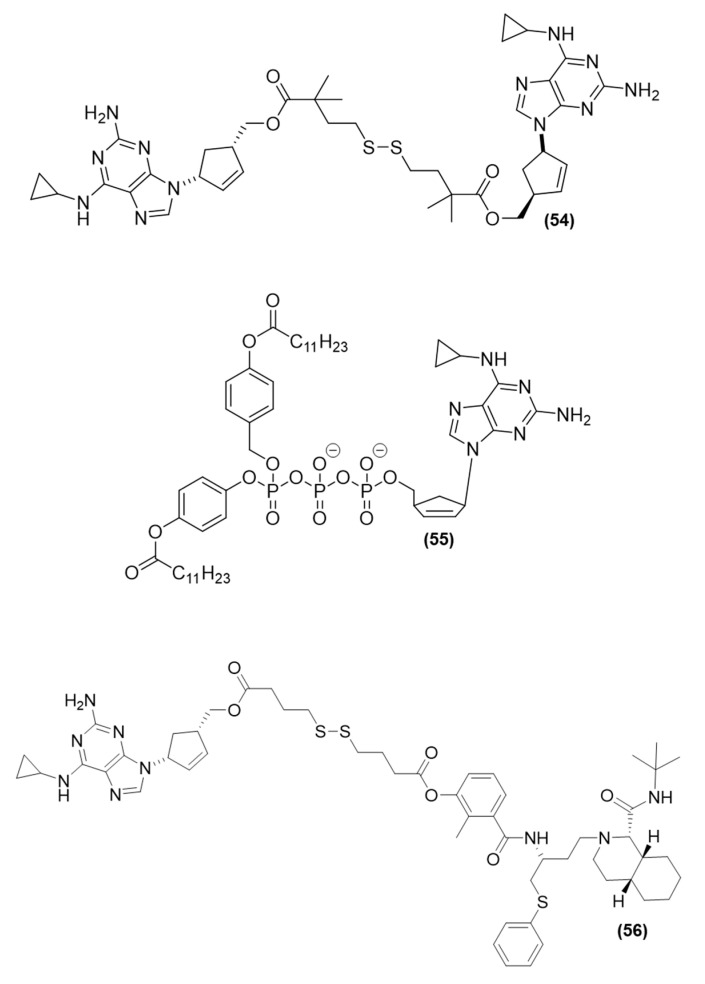
Structural representation of abacavir prodrugs.

## Data Availability

Not applicable.

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
