# Peer review of "The Application of Prodrugs as a Tool to Enhance the Properties of Nucleoside Reverse Transcriptase Inhibitors"

_viruses, 2023, doi:10.3390/v15112234_

Round 1

Reviewer 1 Report

Comments and Suggestions for Authors

Please see review attached.

Author Response

The authors are thankful for the reviewer comments. We agree with all of them. We have corrected and included all the informations requested in this minor review.

Reviewer 2 Report

Comments and Suggestions for Authors

The review by da Rocha Fernandez et al. describes the application of prodrug approaches to enhance properties of NRTIs used for treatment of HIV and other viruses.

General comments

This is a comprehensive review but perhaps too technical; it includes many terms and data that are highly specific to the field of pharmacology/drug discovery and are not explained for a broad audience. To make it more useful to the reader, the authors should consider summarizing after each section which of the approaches described by the different studies cited worked best at enhancing specific properties or addressing issues for the drug discussed. Also, the language used needs review and editing for clarity and to fix errors (please see some examples below)

Specific comments:

The acronym HAART is no longer used; instead, the authors should use antiretroviral therapy (ART).

Line 43: Sentence not clear, please rephrase “emerge in the horizon as perspectives”.

Line 47: The statement “can be used to eliminate undetectable HIV RNA in plasma” should say “detectable”.

Line 111-112: Please add references for Horwitz et al. and Yarchoan et al. In addition, there are other references missing to support the data described or author’ statements.

Line 122: Figure 2 is cited in the text before Figure 1; the authors should re-number the figures so they are in order (Fig 1 should become Fig 2 and vice versa)

Figure 2 should have a legend to orient the reader and briefly explain mechanisms or approaches depicted. The legend should also spell out any abbreviations used. The same comment applies to all the remaining figures.

Lines 131-132: Please define EC50 and CC50

Line 136-137: Please state which are the “other compounds” in this comparison.

Line 159: This sentence seems to be missing a section.

Line 200 and 202: Please define SAR and ID50

Line 207: The term “administering” seems inappropriate in this sentence. Please revise.

Line 254: Sentence “Nasal administration……brain uptake” seems out of place.

Line 286: Please explain “ProTide”

Line 335: Correct word “ediagainst”

Line 394: Remove phrase in parenthesis.

Line 404: Please revise phrase “RNA and DNA are repeatedly verified” as is not clear what this statement means.

Line 408: “within 3-6 days” of which event? Infection? Please clarify.

Line 416: In Figure 4 please correct the English translation of the figure title.

Line 438: Please revise number punctuation. Does 1.000 nM mean one thousand (1,000) nM or 1 nM with no decimals? Also please review and revise units for other measurements as well, such as AUC.

Line 1144: The statement “it might considerably reduce the duration of treatment” seems inaccurate; until a strategy is proven successful at eliminating all HIV reservoirs, it is very likely that HIV infection will continue requiring life-long treatment. 

Comments on the Quality of English Language

The quality of English is in general good, except for a few passages/sentences that seem awkward or used an inappropriate term. These should be revised for clarity. 

Author Response

Dear Reviewer, 

Thanks a lot for your time and efforts to improve the quality of this manuscript. We are in accordance with all of your comments and we have carried out the corrections, according to your sugestions.

General comments

This is a comprehensive review but perhaps too technical; it includes many terms and data that are highly specific to the field of pharmacology/drug discovery and are not explained for a broad audience. To make it more useful to the reader, the authors should consider summarizing after each section which of the approaches described by the different studies cited worked best at enhancing specific properties or addressing issues for the drug discussed. Also, the language used needs review and editing for clarity and to fix errors (please see some examples below)

Answer: We have included some definitions of drug discovery in the text, as for example, the concept of Protide (among otehrs) in order to make it more comprehensive for  a broad audience. We also make a summary after sections to conclude the idea employed. The english editing was submitted by a specialized company responsible for improving the quality. This comment was valuable to improve the text.

Specific comments:

The acronym HAART is no longer used; instead, the authors should use antiretroviral therapy (ART).

Answer: We have corrected through the manuscript.

Line 43: Sentence not clear, please rephrase “emerge in the horizon as perspectives”.

Answer: We have corrected it in the manuscript.

Line 47: The statement “can be used to eliminate undetectable HIV RNA in plasma” should say “detectable”.

Answer: We have corrected it in the manuscript.

Line 111-112: Please add references for Horwitz et al. and Yarchoan et al. In addition, there are other references missing to support the data described or author’ statements.

Answer: We have corrected it in the manuscript and the reference was included.

Line 122: Figure 2 is cited in the text before Figure 1; the authors should re-number the figures so they are in order (Fig 1 should become Fig 2 and vice versa)

Answer: We have corrected it and renumbering in the manuscript.

Figure 2 should have a legend to orient the reader and briefly explain mechanisms or approaches depicted. The legend should also spell out any abbreviations used. The same comment applies to all the remaining figures.

Answer: We have corrected it in the manuscript.

Lines 131-132: Please define EC50 and CC50

Answer: We have included it in the manuscript.

Line 136-137: Please state which are the “other compounds” in this comparison.

Answer: We have corrected it in the manuscript.

Line 159: This sentence seems to be missing a section.

Answer: We have corrected it in the manuscript.

Line 200 and 202: Please define SAR and ID50

Answer: We have corrected it in the manuscript.

Line 207: The term “administering” seems inappropriate in this sentence. Please revise.

Answer: We have corrected it in the manuscript.

Line 254: Sentence “Nasal administration……brain uptake” seems out of place.

Answer: We have corrected it in the manuscript.

Line 286: Please explain “ProTide”

Answer: We have included it in the manuscript.

Line 335: Correct word “ediagainst”

Answer: We have corrected it in the manuscript.

Line 394: Remove phrase in parenthesis.

Answer: We have corrected it in the manuscript.

Line 404: Please revise phrase “RNA and DNA are repeatedly verified” as is not clear what this statement means.

Answer: We have corrected it in the manuscript.

Line 408: “within 3-6 days” of which event? Infection? Please clarify.

Answer: We have corrected it in the manuscript.

Line 416: In Figure 4 please correct the English translation of the figure title.

Line 438: Please revise number punctuation. Does 1.000 nM mean one thousand (1,000) nM or 1 nM with no decimals? Also please review and revise units for other measurements as well, such as AUC.

Answer: We have corrected it in the manuscript.

Line 1144: The statement “it might considerably reduce the duration of treatment” seems inaccurate; until a strategy is proven successful at eliminating all HIV reservoirs, it is very likely that HIV infection will continue requiring life-long treatment. 

Answer: We have corrected it in the manuscript.

Reviewer 3 Report

Comments and Suggestions for Authors

This reviewsummarizes different strategies for obtaining NNRTI prodrugs against HIV in a clear way. However, the article requires some small corrections:

- numbering of compounds: it is confusing that the numbering of the different compounds is not correlative and consistent (in some places they are described only with the name, other times with Arabic numerals and other times with Roman numerals). The compounds should be numbered not with the numbers of the original article but in order of appearance in this review.

- In schemes, compounds should appear in the order in which they appear in the text for ease of reading.

- Throughout the text the names of the compounds (TDF, TFV, etc.) sometimes appear in bold and other times without it, even during the same paragraph (for example: line 840 d4TTP, line 853 d4TTP). Unify all text formatting

- line 324: The text describes the cleavage of ProTide. It should be reflected in the outline for ease of reading.

- line 394: delete the phrase in Portuguese (talvez trocar por "wrap")

- line 739: "CycloSal prodrug [...] (Figure 8)" Cyclosal does not appear in Figure 8. Include the structure.

- line 771 and 776: "D4TDP" (line 771), "D4T" (line 776). Do you meaning d4TDP?. Check the rest of the text anr remove capital D.

- line 1040: remove bold format in NRTI

Author Response

This review summarizes different strategies for obtaining NNRTI prodrugs against HIV in a clear way. However, the article requires some small corrections:

Answer: the authors are thankful for the time you spent reviewing and contributing to improve the quality of the manuscript. We agree with all the comments and contributions and the corrections were done, as requested.

- numbering of compounds: it is confusing that the numbering of the different compounds is not correlative and consistent (in some places they are described only with the name, other times with Arabic numerals and other times with Roman numerals). The compounds should be numbered not with the numbers of the original article but in order of appearance in this review.

Answer: we have corrected the number of all compounds, as requested.

- In schemes, compounds should appear in the order in which they appear in the text for ease of reading.

Answer: we have modified it according to the suggestion. 

- Throughout the text the names of the compounds (TDF, TFV, etc.) sometimes appear in bold and other times without it, even during the same paragraph (for example: line 840 d4TTP, line 853 d4TTP). Unify all text formatting

Answer: we have corrected it according to the suggestion. 

- line 324: The text describes the cleavage of ProTide. It should be reflected in the outline for ease of reading.

Answer: we have modified it according to the suggestion. 

- line 394: delete the phrase in Portuguese (talvez trocar por "wrap")

Answer: we have modified it according to the suggestion. 

- line 739: "CycloSal prodrug [...] (Figure 8)" Cyclosal does not appear in Figure 8. Include the structure.

Answer: we have included it according to the suggestion. 

- line 771 and 776: "D4TDP" (line 771), "D4T" (line 776). Do you meaning d4TDP?. Check the rest of the text anr remove capital D.

Answer: we have modified it according to the suggestion. 

- line 1040: remove bold format in NRTI

Answer: we have modified it according to the suggestion. 

Reviewer 4 Report

Comments and Suggestions for Authors

Santos et al. discussed the use of nucleoside reverse transcriptase inhibitor prodrugs in the treatment of HIV infection (or HBV infection also?). However, low awareness of authors of the topic of HIV and drugs for HIV infection treatment makes this review superficial. The review is poorly written, contains many grammatical flaws and lacks correct citation. The review is also poorly structured and contains a lot of unnecessary and boring information – it is very difficult to read and grasp important information. A small amount of necessary information from the cited articles is not discussed in any way, just pulled from the cited articles. Based on this, I cannot support the publication of the review in this Q1 journal.

General questions and commentaries:

1.       Please provide a critical discussion of the previous results.

2.       Please use last name of the corresponding author or group leader rather than “Some researchers”, or “In a study”, or “Other researchers”, “A research group”. It is very hard to follow which research group you mention.

3.       Please introduce the proper reference in each first sentence, not at the end of the paragraph.

4.       Please focus on HIV rather than other viruses, or rewrite the Abstract and Introduction.

Text-specific questions and commentaries:

Line 5: What does two asterisks ** mean?

Lines 12-14: “The effective treatment of human immunodeficiency virus (HIV), also known as acquired immunodeficiency syndrome (AIDS), has transformed a highly lethal disease into a chronic and manageable condition”. Treatment exists only for infections caused by the virus, not for the virus itself. HIV is a retrovirus; AIDS is a specific condition of HIV infection without treatment. Please re-write this sentence more clearly. 

Line 15: Which reservoirs do you mean?

Lines 31-32: “Considerable advancements in the reduction of mortality and morbidity related to Human Immunodeficiency Virus (HIV) were only possible after the development of the Highly Active Antiretroviral Therapy (HAART)”. … Related to infections caused by HIV…

In addition, uppercase letters does not need here.

Line 33: Reference is needed.

Lines 36-37: “…the discovery of a safe and efficacious therapeutic approach are big problems that need to be addressed”. Isn’t HAART enough?

Line 38: “…only 75% of them have access to HAART”. Only?

Line 39: Please replace “…distributed in…” by “belonging to”

Line 40: Reference is needed.

Line 41: Reference is needed.

Line 46: “Combined therapy” Do you mean HAART?

Line 46: “…at least 2–3 different antiretroviral drugs…” Such therapy contains not just 2-3 different antiretroviral drugs, but antiretroviral drugs from 2-3 different classes.

Lien 47: Reference is needed.

Line 56: Reference is needed.

Line 64: “Among the strategies to overcome the limitations of antiviral drugs, prodrugs might be administered to modulate pharmacokinetic properties” Do antiretroviral agents have poor pharmacokinetics? The introduction discusses a completely different set of issues related to antiretroviral therapy and there is no information on the pharmacokinetics of available antiretroviral agents. So it is not clear to me why prodrugs should be used in antiretroviral therapy. An introduction to the topic of the review should be written.

Line 83-96: The text is very difficult to follow without a figure. There is a lot of deep structural information that does not make any sense.

Line 84: Reference is urgently needed.

Line 98: What is that “RT”? Reverse transcriptase? The abbreviation should be entered into the text before it is used.

Line 99: Reference is needed.

Lines 111-112: References are missing.

Line 113: The use of zidovudine is not discontinued. Moreover, ZDV is still on the WHO List of Essential Medicines.

Lines 113-114: “…bone marrow toxicity, neutropenia, macrocytic anemia, and granulocytopenia”. This called adverse effects.

Line 119: Reference is urgently needed.

Line 129: Reference is needed.

Lines 133, 134: The selectivity index means nothing. The main goal of prodrugs is to improve pharmacokinetics, as you stated.

Line 139: “In a study…” Please, use the last name of the corresponding author. 

Line 153: Reference is needed.

Line 160: Remove " between “(AZT).” and “The..”

Line 193, 194: References are missing.

Lines 233-238 Please avoid these specific descriptions from the original article, but provide your own critical statement of the previous findings.

Line 245 “the same research group” Which? Please introduce this group first using the last name of group leader.

Line 265: Reference is needed.

Lines 265-268: Please re-write these sentences in another way.

Line 275: “… in treating naïve patients”. What do you mean by that? Healthy volunteers? Children? I don’t get this.

Line 286: What is ProTide?

Line 302: “In clinical trials, TAF was well-tolerated and presented potent antiviral activity”. The term “activity” is associated with in vitro studies. Therapeutic effect in humans or animals is commonly referred as “efficacy”.

Line 303: “About 8 mg of TAF had antiviral effects similar to that of 300 mg of TFV-DP”. What is meant by “antiviral effect”? Each clinical trials contains trial-specific clinical endpoints that may or may not be met.

Line 399: “This type of transport can avoid a first-pass effect.” Which?

Line 419: Reference is needed.

Line 424: Reference is needed.

Line 424: “…for the 27 different viruses evaluated…” These are not different viruses, these are different isolates and strains of the same virus of interest, HIV.

Line 502: 0.00050 μM is 0.5 nM

Line 516: Reference is needed.

Line 521: Reference is needed.

Lines 544-554: The topic of your review is the “Application of Prodrugs as a Tool to Enhance the Properties of Nucleoside Reverse Transcriptase Inhibitors for HIV”, not hepatitis B virus.

Line 557: Reference is needed.

Line 573: Reference is needed.

Line 585: Reference is needed.

Line 587: What does “HepDirect” mean? Is this the commercial name of technology?

Lien 634: Reference is needed.

Line 647: Reference is needed.

Line 663: I don’t clearly understand now, what is your main goal of the review? HIV or HBV infections?

Line 675: Reference is needed.

Line 686: Reference is needed.

Line 699: Reference is needed.

Line 717: Reference is needed.

Line 740: Reference is needed.

Line 747: Reference is needed.

Line 758: Reference is needed.

Line 758: “Two different masking units” Which?

Line 761: “…using phosphate buffer, RPMI culture medium, CEM/0 cell extracts, and pig liver esterase”. Please avoid unnecessary details. All of these words might be replaced by “different experimental conditions” or something similar.

Line 784: Reference is needed.

Line 802: Reference is needed.

Line 819: Reference is needed.

Line 855: What does “TriPPPro” mean?

Line 857: Reference is needed.

Line 862: Reference is needed.

Line 889: Reference is needed.

Line 908: Reference is needed.

Line 919: Reference is needed.

Line 921: Reference is needed.

Line 938: Reference is needed.

Line 908: “In vitro evaluations of…” Do you mean “In vitro ADME”?

Line 949: Reference is needed.

Line 966: “CCRF-CEM, ATCC No.CCL-119” Why this information here?

Line 970-971 “The HPLC profiles also revealed peaks at 1.7–2.9 min, indicating that intracellular hydrolysis occurred and potentially phosphorylated products were present.” What molecular masses have peaks at retention times 1.7-2.9? The peaks themselves mean nothing and do not indicate anything. Mass detection is needed to describe the observed molecular masses for degradation products. The sentence is technically incorrect and should be re-written.

Line 974: Reference is needed.

Line 998: Why is the central nervous system a critical location? Are other systems not critical?

Line 1000: Reference is needed.

Line 1027: Reference is needed.

Line 1042: What do you mean by “a Trojan horse strategy”?

Line1043: Reference is needed.

Line1073: Reference is needed.

Line 1073-1074 This sentence repeats the first sentence but with different words, please delete or rewrite it.

Line 1083: Reference is needed.

Line 1128 This section does not contain any conclusion or outlook of the review. It seems like Abstract. Please completely rewrite it.

Comments on the Quality of English Language

Extensive editing of English language required.

Author Response

Santos et al. discussed the use of nucleoside reverse transcriptase inhibitor prodrugs in the treatment of HIV infection (or HBV infection also?). However, low awareness of authors of the topic of HIV and drugs for HIV infection treatment makes this review superficial. The review is poorly written, contains many grammatical flaws and lacks correct citation. The review is also poorly structured and contains a lot of unnecessary and boring information – it is very difficult to read and grasp important information. A small amount of necessary information from the cited articles is not discussed in any way, just pulled from the cited articles. Based on this, I cannot support the publication of the review in this Q1 journal.

Answer: the authors are thankful for the comments and insights provide. The manuscript review an important approach used in medicinal chemistry to improve the efficacious and safety of new drugs - the prodrug approach. Its use to drug discovery in for the HIV arsenal is very rich and not explored in the  literature. This is a innovative paper in such way, and the aim was to provide to all readers of this important journal how chemistry is contributing for HIV drug discovery using successful examples. Based on the reviewer comments we clarified some points in the text according his suggestion, and we thank for the contrinution for this second version. 

General questions and commentaries:

  1. Please provide a critical discussion of the previous results.

Answer: We have included it as suggested.

  1. Please use last name of the corresponding author or group leader rather than “Some researchers”, or “In a study”, or “Other researchers”, “A research group”. It is very hard to follow which research group you mention.

Answer: We have done it as suggested

  1. Please introduce the proper reference in each first sentence, not at the end of the paragraph.

Answer: We have included it as suggested.

  1. Please focus on HIV rather than other viruses, or rewrite the Abstract and Introduction.

Answer: We have included it as suggested. As expert in the field, the reviewer knows that some drugs used to treat HIV is also used to treat other viral infections. This is the reason we have in some moments a superficial comment about the application of the prodrug to other virus. In this version, we tried to remove unnecessary notes about other virus to improve the comprehension and avoid confusion by all readers. We thank the reviewer for this observation. All corrections were done.

Text-specific questions and commentaries:

Line 5: What does two asterisks ** mean?

Answer: It means that  the email jean.santos@unesp.br is the correspondence.

Lines 12-14: “The effective treatment of human immunodeficiency virus (HIV), also known as acquired immunodeficiency syndrome (AIDS), has transformed a highly lethal disease into a chronic and manageable condition”. Treatment exists only for infections caused by the virus, not for the virus itself. HIV is a retrovirus; AIDS is a specific condition of HIV infection without treatment. Please re-write this sentence more clearly. 

Answer: We have corrected it as suggested. The statement now is: "he Antiretroviral Therapy (ART) is an effective treatment of Human Immunodeficiency Virus (HIV) which has transformed the highly lethal disease, Acquired Immunodeficiency Syndrome (AIDS), into a chronic and manageable condition".

Line 15: Which reservoirs do you mean?

Answer: The reservoir for HIV in different tissues. We included more details to avoid confusion.

Lines 31-32: “Considerable advancements in the reduction of mortality and morbidity related to Human Immunodeficiency Virus (HIV) were only possible after the development of the Highly Active Antiretroviral Therapy (HAART)”. … Related to infections caused by HIV…

Answer: We have corrected it as suggested.

In addition, uppercase letters does not need here.

Answer: We have corrected it as suggested.

Line 33: Reference is needed.

Answer: We have included it as suggested.

Lines 36-37: “…the discovery of a safe and efficacious therapeutic approach are big problems that need to be addressed”. Isn’t HAART enough?

Answer: No, ART is efficacious for the majority of patients but some of them, mainly in developing countries, the lack of the best options and the developing of resistance are challenges faced daily. In addition, we are living in a second moment, in which the mitigation of reservoir, for example, is an approach pursued by those who work in the drug discovery. Other strategies beyond ART is necessary to advance in the field.

Line 38: “…only 75% of them have access to HAART”. Only?

Answer: Yes, only 75%; bacause it is important to note that those 25% that are not receving ART treatment is transmitting and possibly developing resistant strains. The WHO goal is to provide 100% of treatment for all patients. This is the reason that 75% is low, far from the established aims. 

Line 39: Please replace “…distributed in…” by “belonging to”

Answer: We have corrected it as suggested.

Line 40: Reference is needed.

Answer: We have included it as suggested.

Line 41: Reference is needed.

Answer: We have included it as suggested.

Line 46: “Combined therapy” Do you mean HAART?

Answer: yes.

Line 46: “…at least 2–3 different antiretroviral drugs…” Such therapy contains not just 2-3 different antiretroviral drugs, but antiretroviral drugs from 2-3 different classes.

Answer: Not always. One usual scheme we used here in the practice is the combination of lamivudine + tenofovir (both nucleoside reverse transcriptase inhibitor) and dolutegravir. 

Lien 47: Reference is needed.

Answer: We have included it as suggested.

Line 56: Reference is needed.

Answer: We have included it as suggested.

Line 64: “Among the strategies to overcome the limitations of antiviral drugs, prodrugs might be administered to modulate pharmacokinetic properties” Do antiretroviral agents have poor pharmacokinetics? The introduction discusses a completely different set of issues related to antiretroviral therapy and there is no information on the pharmacokinetics of available antiretroviral agents. So it is not clear to me why prodrugs should be used in antiretroviral therapy. An introduction to the topic of the review should be written.

Answer: We have included it as suggested.

Line 83-96: The text is very difficult to follow without a figure. There is a lot of deep structural information that does not make any sense.

Answer: We have corrected it as suggested.

Line 84: Reference is urgently needed.

Answer: We have included it as suggested.

Line 98: What is that “RT”? Reverse transcriptase? The abbreviation should be entered into the text before it is used.

Answer: We have included it as suggested.

Line 99: Reference is needed.

Answer: We have included it as suggested.

Lines 111-112: References are missing.

Answer: We have included it as suggested.

Line 113: The use of zidovudine is not discontinued. Moreover, ZDV is still on the WHO List of Essential Medicines.

Answer: We have corrected it as suggested. Yes, ZDV is still on the WHO list. 

Lines 113-114: “…bone marrow toxicity, neutropenia, macrocytic anemia, and granulocytopenia”. This called adverse effects.

Answer: We have included it as suggested.

Line 119: Reference is urgently needed.

Answer: We have included it as suggested.

Line 129: Reference is needed.

Answer: We have included it as suggested.

Lines 133, 134: The selectivity index means nothing. The main goal of prodrugs is to improve pharmacokinetics, as you stated.

Answer: We have corrected it as suggested.

Line 139: “In a study…” Please, use the last name of the corresponding author. 

Answer: We have included it as suggested.

Line 153: Reference is needed.

Answer: We have included it as suggested.

Line 160: Remove " between “(AZT).” and “The..”

Answer: We have included it as suggested.

Line 193, 194: References are missing.

Answer: We have included it as suggested.

Lines 233-238 Please avoid these specific descriptions from the original article, but provide your own critical statement of the previous findings.

Answer: We have included it as suggested.

Line 245 “the same research group” Which? Please introduce this group first using the last name of group leader.

Answer: We have included it as suggested.

Line 265: Reference is needed.

Answer: We have included it as suggested.

Lines 265-268: Please re-write these sentences in another way.

Answer: We have included it as suggested.

Line 275: “… in treating naïve patients”. What do you mean by that? Healthy volunteers? Children? I don’t get this.

Answer: We have included it as suggested.

Line 286: What is ProTide?

Answer: We have included it as suggested.

Line 302: “In clinical trials, TAF was well-tolerated and presented potent antiviral activity”. The term “activity” is associated with in vitro studies. Therapeutic effect in humans or animals is commonly referred as “efficacy”.

Answer: We have corrected it as suggested.

Line 303: “About 8 mg of TAF had antiviral effects similar to that of 300 mg of TFV-DP”. What is meant by “antiviral effect”? Each clinical trials contains trial-specific clinical endpoints that may or may not be met.

Answer: We have included it as suggested. 

Line 399: “This type of transport can avoid a first-pass effect.” Which?

Answer: We have included it as suggested. The correct is 'lymphatic' transport.

Line 419: Reference is needed.

Answer: We have included it as suggested. 

Line 424: Reference is needed.

Answer: We have included it as suggested. 

Line 424: “…for the 27 different viruses evaluated…” These are not different viruses, these are different isolates and strains of the same virus of interest, HIV.

Answer: We have corrected it as suggested. 

Line 502: 0.00050 μM is 0.5 nM

Answer: We have corrected it as suggested.

Line 516: Reference is needed.

Answer: We have included it as suggested.

Line 521: Reference is needed.

Answer: We have included it as suggested.

Lines 544-554: The topic of your review is the “Application of Prodrugs as a Tool to Enhance the Properties of Nucleoside Reverse Transcriptase Inhibitors for HIV”, not hepatitis B virus.

Answer: Yes, in some studies the authors evaluated in both HIV and hepatitis. But we agree with the reviewer and we only comment in a superficial way about the use in hepatitis B

Line 557: Reference is needed.

Answer: We have included it as suggested.

Line 573: Reference is needed.

Answer: We have included it as suggested.

Line 585: Reference is needed.

Answer: We have included it as suggested.

Line 587: What does “HepDirect” mean? Is this the commercial name of technology?

Answer: We have included it as suggested. HepDirect means liver-targeting approach .

Lien 634: Reference is needed.

Answer: We have included it as suggested.

Line 647: Reference is needed.

Answer: We have included it as suggested.

Line 675: Reference is needed.

Answer: We have included it as suggested.

Line 686: Reference is needed.

Answer: We have included it as suggested.

Line 699: Reference is needed.

Answer: We have included it as suggested.

Line 717: Reference is needed.

Answer: We have included it as suggested.

Line 740: Reference is needed.

Answer: We have included it as suggested.

Line 747: Reference is needed.

Answer: We have included it as suggested.

Line 758: Reference is needed.

Answer: We have included it as suggested.

Line 758: “Two different masking units” Which?

Answer: We have included it as suggested.

Line 761: “…using phosphate buffer, RPMI culture medium, CEM/0 cell extracts, and pig liver esterase”. Please avoid unnecessary details. All of these words might be replaced by “different experimental conditions” or something similar.

Answer: We have included it as suggested.

Line 784: Reference is needed.

Answer: We have included it as suggested.

Line 802: Reference is needed.

Answer: We have included it as suggested.

Line 819: Reference is needed.

Answer: We have included it as suggested.

Line 855: What does “TriPPPro” mean?

Answer: We have included it as suggested. It means nucleoside triphosphate prodrugs (TriPPPro) – approaches 

Line 857: Reference is needed.

Answer: We have included it as suggested.

Line 862: Reference is needed.

Answer: We have included it as suggested.

Line 889: Reference is needed.

Answer: We have included it as suggested.

Line 908: Reference is needed.

Answer: We have included it as suggested.

Line 919: Reference is needed.

Answer: We have included it as suggested.

Line 921: Reference is needed.

Answer: We have included it as suggested.

Line 938: Reference is needed.

Answer: We have included it as suggested.

Line 908: “In vitro evaluations of…” Do you mean “In vitro ADME”?

Answer: Yes.

Line 949: Reference is needed.

Answer: We have included it as suggested.

Line 966: “CCRF-CEM, ATCC No.CCL-119” Why this information here?

Answer: We have removed it as suggested.

Line 970-971 “The HPLC profiles also revealed peaks at 1.7–2.9 min, indicating that intracellular hydrolysis occurred and potentially phosphorylated products were present.” What molecular masses have peaks at retention times 1.7-2.9? The peaks themselves mean nothing and do not indicate anything. Mass detection is needed to describe the observed molecular masses for degradation products. The sentence is technically incorrect and should be re-written.

Answer: We have included it as suggested. The two peaks indicates the formation of distinct products after hydrolysis. We reformulate the statement to avoid confusion, as follow: "The HPLC profiles also indicate that intracellular hydrolysis occurred". 

Line 974: Reference is needed.

Answer: We have included it as suggested.

Line 998: Why is the central nervous system a critical location? Are other systems not critical?

Answer: Its a critical location to be accessed by small molecules (drugs) due to its characteristics.

Line 1000: Reference is needed.

Answer: We have included it as suggested.

Line 1027: Reference is needed.

Answer: We have included it as suggested.

Line 1042: What do you mean by “a Trojan horse strategy”?

Answer: We have changed for prodrug strategy.

Line1043: Reference is needed.

Answer: We have included it as suggested.

Line1073: Reference is needed.

Answer: We have included it as suggested.

Line 1073-1074 This sentence repeats the first sentence but with different words, please delete or rewrite it.

Answer: We have deleted it as suggested.

Line 1083: Reference is needed.

Answer: We have included it as suggested.

Line 1128 This section does not contain any conclusion or outlook of the review. It seems like Abstract. Please completely rewrite it.

Answer: We have included it as suggested.
